# 3D Question Answering via only 2D Vision-Language Models

**Fengyun Wang** [1]  **Sicheng Yu** [2]  **Jiawei Wu** [3]  **Jinhui Tang** [4]  **Hanwang Zhang** [1]  **Qianru Sun** [2]

## Abstract

Large vision-language models (LVLMs) have significantly advanced numerous fields. In this work, we explore how to harness their potential to address 3D scene understanding tasks, using 3D question answering (3D-QA) as a representative example. Due to the limited training data in 3D, we do not train LVLMs but infer in a zero-shot manner. Specifically, we sample 2D views from a 3D point cloud and feed them into 2D models to answer a given question. When the 2D model is chosen, e.g., LLAVA-OV, the quality of sampled views matters the most. We propose `cdViews`, a novel approach to automatically selecting critical and diverse `Views` for 3D-QA. `cdViews` consists of two key components: `viewSelector` prioritizing critical views based on their potential to provide answer-specific information, and `viewNMS` enhancing diversity by removing redundant views based on spatial overlap. We evaluate `cdViews` on the widely-used ScanQA and SQA benchmarks, demonstrating that it achieves state-of-the-art performance in 3D-QA while relying solely on 2D models without fine-tuning. These findings support our belief that 2D LVLMs are currently the most effective alternative (of the resource-intensive 3D LVLMs) for addressing 3D tasks. *The code is available at* `https://github.com/fereenwong/cdViews`.

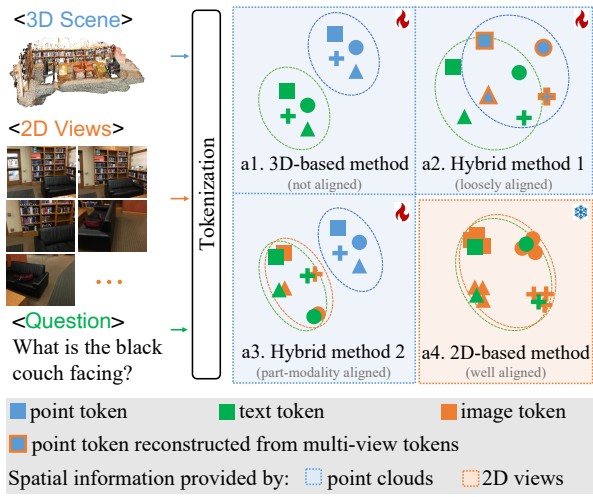

(a) Illustration of Feature Alignment Issue

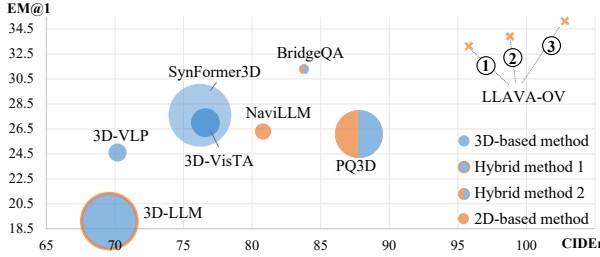

(b) Performance on the test set (with objects) of ScanQA

Figure 1: **Comparison of 3D Question Answering methods**. **(a):** a1 for 3D-based methods; a2 and a3 for hybrid (2D+3D) methods. All of these methods require computationally intensive 3D-language alignment using point cloud data for spatial reasoning. a4 is our method that leverages pre-trained LVLMs operating solely on 2D views. The well-aligned features between 2D visual features and language in 2D LVLMs enable zero-shot 3D-QA. **(b):** Model comparison on the test set (with objects) of ScanQA. The upper-right corner indicates the best performance. The circle area represents the size of training data required for aligning 3D and language. The "✕" denotes zero-shot 3D-QA using 2D model LLAVA-OV (Li et al., 2024a). We respectively use ① uniform sampling, ② image retrieval, and ③ our `cdViews`, to select views as input to LLAVA-OV.

## 1. Introduction

The advancement of large vision-language models (LVLMs) has transformed the vision-language domain by jointly processing huge sets of vision and text training data, leading to

---
[*]Equal contribution  [1]Nanyang Technological University, Singapore [2]Singapore Management University, Singapore [3]National University of Singapore, Singapore [4]Nanjing University of Science & Technology, Nanjing, China. Correspondence to: Qianru Sun <qianrusun@smu.edu.sg>.

*Proceedings of the $42^{nd}$ International Conference on Machine Learning*, Vancouver, Canada. PMLR 267, 2025. Copyright 2025 by the author(s).

significant breakthroughs in addressing 2D visual question

answering (2D-VQA) (Shao et al., 2023; Guo et al., 2023; Lu et al., 2023). However, extending these capabilities to 3D question answering (3D-QA) has unique challenges. Unlike 2D tasks, which benefit from abundant paired training data, the 3D domain lacks large-scale datasets to learn the alignment between 3D (such as point clouds) and language (such as text descriptions of 3D scenes). Existing 3D-language models still fall short of serving as robust counterparts to the widely used 2D-language models such as LLaMA-3 (Dubey et al., 2024). Therefore, current 3D-QA methods often have to train from scratch on small-scale 3D datasets, resulting in poor model performance. In contrast, hybrid approaches leverage additional 2D information. One solution (Hong et al., 2023) is to reconstruct 3D features from the features of multiple 2D views (Figure 1 (a2)), but its performance is poor due to the technical challenge of 3D reconstruction. Another solution (Mo & Liu, 2024) is to combine 2D and 3D features as input into the model (Figure 1 (a3)). 2D features extracted from LVLMs are already well-aligned with language, but further alignment with 3D features requires careful model design and advanced training techniques. Figure 1(b) shows that hybrid methods also require extensive amounts of training data (indicated by the large circle area), which are not always available.

In this paper, we take a completely different approach by avoiding direct alignment between 3D and language. Instead, we rely solely on 2D views and pre-trained LVLMs for understanding 3D scenes. For implementation, we first select a limited number of 2D views, and then take them as the only visual input to LVLMs to answer the input question.

During our preliminary trials, we identified several challenges. First, all LVLMs have a token limit, restricting the number of 2D views they can process at once. This constraint makes it crucial to carefully select the most informative views. Second, given a fixed number of views, the quality of the selected views plays a critical role. Existing methods for view selection fall into two categories: uniform sampling, which randomly selects views, and image retrieval, which selects views based on question-based retrieval (Li et al., 2022). However, both approaches have significant limitations, either being inefficient or failing to capture critical views. Specifically, as shown in Figure 2, image retrieval outperforms uniform sampling but has two major limitations. First, it prioritizes question-related views over truly essential ones for answering the question. For example, when asked "What is the black couch facing?", the model retrieves images of the "couch" but overlooks the "coffee table", which is the answer-related object but in the opposite view of "couch". Second, it often selects redundant or overlapping views, causing inefficiency.

To tackle the challenges, we introduce a new framework `cdViews` to select critical and diverse Views

\<Question\>: What is the black couch facing?
\<Answer\>: Coffee table

**Uniform Sampling** -- ignores question context

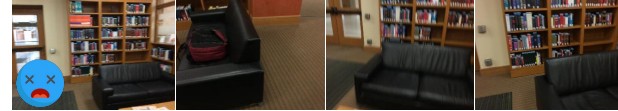

**Image Retrieval** – overlooks answer-related information

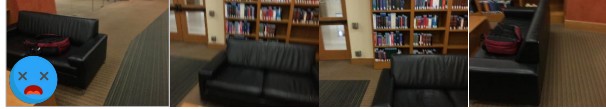

**Ours** – "the black couch facing a coffee table" is included

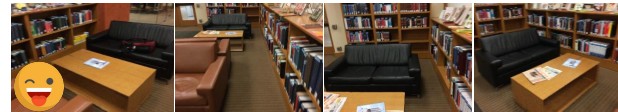

Figure 2: **Comparison of view selection methods.**

(`cdViews`) and then use them to perform LVLMs-based 3D-QA in a zero-shot manner. `cdViews` is designed on two key principles. (1) **Prioritize Critical Views**: We aim for views that contain information crucial for answering questions, rather than merely finding views that match question texts. Thus, we develop a lightweight `viewSelector` module that prioritizes views most likely to contain answer-related information. To train this module, we design a `viewAnnotator` that automatically generates training data in two steps. `viewAnnotator` firstly converts question-answer pairs into descriptive captions. It then leverages a pre-trained LVLM to identify the most informative views that match these captions. (2) **Enhance View Diversity**: The aim is to improve spatial diversity and minimize redundancy for the selected views. To this end, we develop a view Non-Maximum Suppression method dubbed as `viewNMS`. This method uses camera parameters, including position and orientation, to filter out overlapping views while preserving spatial views as diverse as possible. When `viewSelector` and `viewNMS` are ready, they will be plugged into a pre-trained 2D LVLM for zero-shot 3D-QA in the inference stage.

We evaluate the proposed `cdViews` on two widely used benchmarks of 3D-QA: ScanQA (Azuma et al., 2022) and SQA (Ma et al., 2022). Our experimental results demonstrate that `cdViews`'s view selection significantly outperforms conventional approaches such as uniform sampling and image-text retrieval. Notably, `cdViews` achieves superior performance compared to models using 3D or hybrid input data. In summary, our contributions are three-fold. (1) We explore the use of 2D-only LVLM to address 3D-QA in a zero-shot manner, analyzing various view selection methods. (2) We introduce `cdViews` that integrates a `viewSelector` with a `viewNMS` to capture critical and diverse views. We design a `viewAnnotator` to generate

training data for `viewSelector` automatically. (3) Our experiment results demonstrate that `cdViews` achieves state-of-the-art performance on two 3D-QA benchmarks, even surpassing the 3D or hybrid models.

## 2. Related Works

Existing approaches to 3D-QA can be categorized into three folds based on the format of visual inputs: 3D-based, 2D-based, and hybrid (combining 3D and 2D).

**3D-based Methods**. The 3D-based methods (Man et al., 2024a) use 3D point clouds as visual input, allowing direct processing of point cloud data to understand 3D environments. However, these methods face two challenges. First, the scarcity of 3D-language training data limits its scalability. Efforts such as 3D-VLP (Yang et al., 2024) attempt to mitigate this issue by leveraging large-scale synthetic datasets, and recent works (Zhang et al., 2024; Jin et al., 2023b; Hong et al., 2023; Zhu et al., 2023; Chen et al., 2024b) aim to unify multiple 3D tasks, such as captioning, question answering, and grounding, under a single framework. Second, using an entire 3D scene as input introduces unnecessary information for QA, distracting the model and reducing efficiency. To address this, methods such as SIG3D (Man et al., 2024a) incorporate situational awareness to focus on only relevant 3D regions guided by the language prompts (e.g., the input situation). Overall, 3D-based methods have constraints due to the lack of large-scale 3D language pretraining data. The resulting 3D-language alignment in the feature space is thus suboptimal. Besides, using entire scenes as input to answer local questions is costly and inefficient.

**2D-based Methods**. Recent 2D-based methods use uniformly sampled 2D views as input to 2D LVLMs (Singh et al., 2024; Zheng et al., 2024; Liu et al., 2024b), primarily focusing on evaluating the performance of 2D LVLMs on 3D-QA. They focus more on evaluating pretrained 2D LVLMs on 3D-QA tasks, rather than developing approaches to adapt and improve their performance for spatial reasoning. Some more recent works have attempted to utilize 2D views more effectively. OpenEQA (Majumdar et al., 2024), transforms visual information into textual context, such as frame-level or scene-graph captions, and then leverages LLMs to answer questions. This approach depends on whether the generated text description can accurately capture the critical visual details, which may lead to incomplete or inaccurate information.

Compared to the above methods, we make two key contributions. First, we are the first to leverage 2D LVLMs via zero-shot inference (or by plugging a lightweight module) to address 3D-QA tasks. Second, we identify view selection as a critical factor in zero-shot 3D-QA, for which there is a

lack of an efficient solution in prior works. To tackle this, we propose a simple yet effective strategy for selecting critical and diverse views (i.e., `cdViews`), thereby enhancing the utility of readily-trained 2D LVLMs for 3D-QA.

**Hybrid Methods**. Hybrid methods (Huang et al., 2024; Mo & Liu, 2024; Huang et al., 2023; Hong et al., 2023; Man et al., 2024b; Fu et al., 2024) leverage pre-trained 2D LVLMs to address 3D vision-language tasks in two main ways. The first approach involves mapping multi-view 2D image features (which are well-aligned with language due to 2D LVLMs) into the 3D feature space (Zhu et al., 2025; Hong et al., 2023). These mapped features can either replace original 3D features (Hong et al., 2023) or serve as complementary inputs to enhance the alignment between language and hybrid (2D+3D) features (Zhu et al., 2025). The second approach processes 2D images and 3D point clouds as parallel inputs (Mo & Liu, 2024), using complementary strengths: 2D views provide fine-grained semantic details, while 3D point clouds capture spatial awareness. Although these methods improve 3D-QA performance, they rely on explicit 3D reconstruction, needing additional models and causing more processing steps. In contrast, our method uses 2D views and feeds them into a unified 2D LVLM, which makes a simpler pipeline.

Different from these hybrid methods, our approach relies solely on 2D views as input, without the need for mapping between 3D and 2D. Our technical contribution is an efficient view selection strategy, `cdViews`. Among the hybrid methods, the work most closely related to ours is BridgeQA (Mo & Liu, 2024), which selects views by first retrieving the top-1 question-related view and then combining it with 3D point clouds as input for a hybrid model. However, BridgeQA depends on 3D point clouds to extract spatial information for QA, requiring complex 3D→2D→language alignment. Additionally, its retrieval-based approach risks overlooking critical views (which we will show in the experimental sections). In contrast, our method leverages multiple 2D views to understand 3D, meanwhile utilizing the strong language alignment already achieved by pre-trained 2D LVLMs.

## 3. Preliminaries

Leveraging pre-trained 2D LVLMs in a zero-shot manner for 3D-QA tasks is promising yet underexplored. Since 2D LVLMs are fundamentally designed to process 2D images as input, we propose `cdViews` to efficiently select the most informative 2D views of 3D scenes. To understand the complexities in view selection, we conduct a preliminary study using intuitive view selection methods, taking LLAVA-OV (Li et al., 2024a) as the backbone and using the validation set of the ScanQA dataset (Azuma et al., 2022). Note that it requires no training data due to the zero-shot

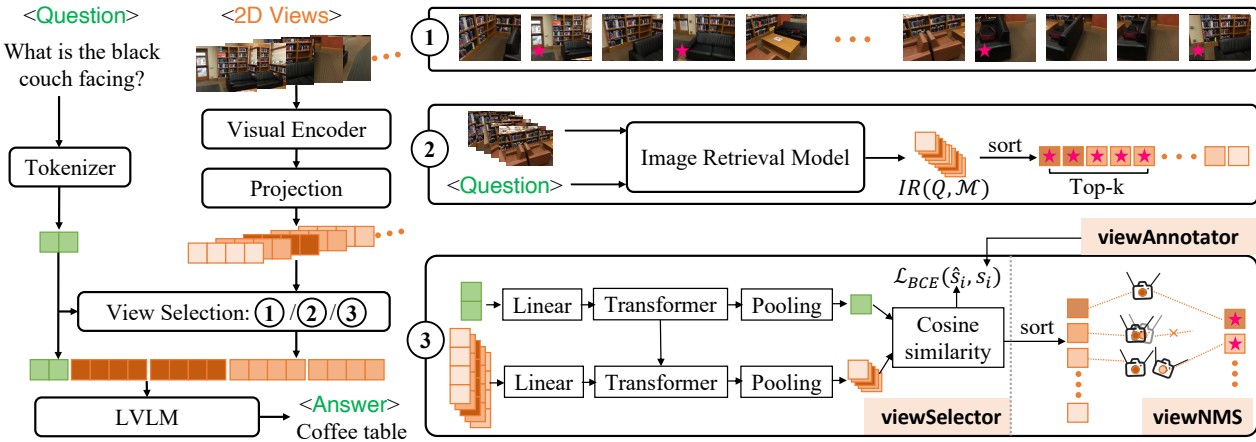

Figure 3: **The pipeline of zero-shot 3D-QA using three different view selection methods: uniform sampling (option ①), image retrieval (option ②), and our `cdViews` (option ③).** The views marked with ★ are selected ones. As for inference, our `cdViews` has two modules to run: the `viewSelector` identifies critical views, and the `viewNMS` enhances view diversity and minimizes redundancy. The `viewSelector` is trained using automatically generated labels from the `viewAnnotator` module, which is detailed in Figure 5.

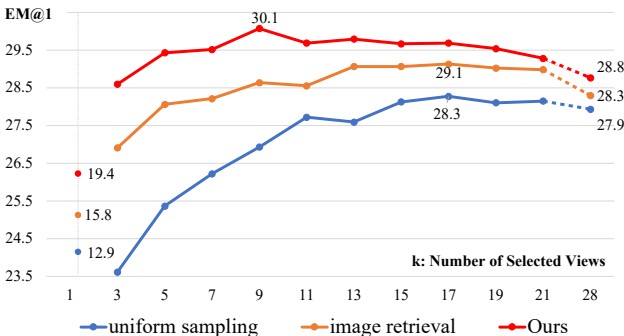

Figure 4: **Performance comparison of view selection methods** on the validation set of ScanQA (Azuma et al., 2022). It can be observed that: 1) performance improves with an increasing number of views, peaks at a certain point, and finally declines; and 2) noticeable performance gaps arise from different view selection methods, highlighting the importance of effective view selection. An earlier peak (30.1) appears in `cdViews` thanks to `viewNMS`.

approach. In the following, we first present a problem formulation for zero-shot 3D-QA, followed by experiments using two intuitive view selection methods: uniform sampling and image retrieval.

**Problem Formulation**. Given a question $Q$ and a 3D scene represented by a set of 2D views $\mathcal{M} = \{V_1, V_2, \ldots, V_N\}$, each associated with a camera matrix containing the position and orientation. The view selection identifies a subset of $k$ views (that are useful to answer $Q$), denoted as $\mathcal{M}'$:

$$\mathcal{M}' = \mathcal{F}(\mathcal{M}, Q, k) = \{V_{i_1}, V_{i_2}, \ldots, V_{i_k}\}, \quad (1)$$

where $k \leq N$, $\mathcal{F}$ is a view selection function, which can

either be question-dependent (denoted as $\mathcal{F}(\mathcal{M}, Q, k)$), or not (denoted as $\mathcal{F}(\mathcal{M}, k)$). Then, $\mathcal{M}'$ and the question $Q$ are input into the model to produce the answer $A$:

$$A = \text{LVLM}(\mathcal{M}', Q). \quad (2)$$

The same zero-shot inference process is applied throughout all experiments in this work, with variations only in two key aspects: the view selection function $\mathcal{F}$ and the number of selected views $k$ that determine the final set of views $\mathcal{M}'$.

**Uniform Sampling *vs.* Image Retrieval**. We show the zero-shot experimental results of these two methods in Figure 4. We also include the results of our `cdViews` for comparison.

1) Uniform sampling randomly selects 2D views without considering the context of the question $Q$ (option ① in Figure 3), formulated as:

$$\mathcal{F}_{\text{uniform}}(\mathcal{M}, k) = \{V_{i_j}\}_{j=1}^k, i_j \sim \text{Uniform}(1, N). \quad (3)$$

Uniform sampling is the most straightforward way to select 2D views as input into 2D LVLMs for 3D-QA, and the best achieved metric score of EM@1 is 28.3%.

2) Image retrieval has been used in BridgeQA (Mo & Liu, 2024). Following (Mo & Liu, 2024), we use the BLIP's image-text retrieval model (Li et al., 2022) to select views that best match the question $Q$ (option ② in Figure 3). This process can be represented as:

$$\mathcal{F}_{\text{retrieval}}(\mathcal{M}, Q, k) = \{V_{i_j} \mid i_j \in \text{Top-}k(\text{IR}(Q, \mathcal{M}))\}. \quad (4)$$

where $\text{IR}(Q, \mathcal{M})$ denotes the semantic similarity scores between $Q$ and every view in $\mathcal{M}$, *i.e.*, identifying the views

**Step 1: Caption Generation**

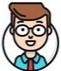

$<Prompt_R>$: You are a helpful assistant. For each QA pair, generate a caption that describes the visual scene, fully incorporating relevant information from the question and answer.

$<$Question$>$: What is in the right corner of room by curtains? $<$Answer$>$: brown cabinet with tv sitting in it

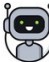

a brown cabinet with a television inside is located in the right corner of the room, near the curtains.

**Step 2: View Matching**

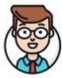

$<Prompt_M>$: You are given an image and a caption describing the visual content. Determine if the image matches the caption, and respond with one of the following options:
A. Yes, fully matches.    B. No, does not match.    C. Uncertain, insufficient or unclear information.

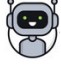

Positive | Negative

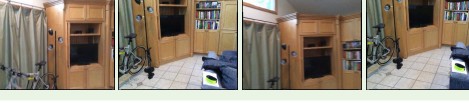 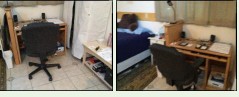 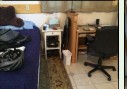 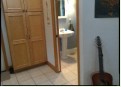

Figure 5: **Our `viewAnnotator`** module operates in two steps: Caption Generation and View Matching (illustrated by light green boxes indicating outputs at each step). In Step 1, LVLMs processes question-answer pairs to produce detailed descriptive captions. In Step 2, these captions are compared against sampled views to assess their relevance in answering the corresponding questions. For clarity, the figure depicts only positive (A) and negative (B) view matches, excluding uncertain (C) ones.

semantically aligned with the question. As shown in Figure 4, the best EM@1 score that this approach achieves is 29.1%, slightly outperforming uniform sampling.

**Analysis**. Overall, image retrieval shows modest improvements over uniform sampling. It relies on the semantic similarity between questions and views, which introduces two key limitations: 1) *Missing Critical Views*. While it effectively identifies views containing objects explicitly mentioned in the question, it frequently overlooks relational cues essential for answering the question. This limitation stems from the fundamental difference between object identification and relationship comprehension, and the latter requiring stronger understanding capabilities. 2) *Redundancy*. Our analysis shows that views from adjacent viewpoints typically receive similar semantic similarity scores, resulting in the selection of overlapping views. This redundancy limits the diversity of visual information captured across multiple views, reducing the overall effectiveness of the image retrieval approach.

## 4. `cdViews`: Critical and Diverse Views

Based on the above analysis, we argue that effective zero-shot 3D-QA requires identifying views that are both *critical* to represent the key information in the scene and sufficiently *diverse* to cover the scene. To this end, we introduce `cdViews`, *i.e.*, the option ③ in Figure 3. In the inference stage of 3D-QA, `cdViews` loads two modules, `viewSelector` and `viewNMS`. The training of `viewSelector` contains two steps: data annotation and model training. First, we propose an auto

`viewAnnotator` to label views as positive, negative, or uncertain based on their matching scores with the descriptive captions (generated from question-answer pairs). Then, we train `viewSelector` with these labels in a supervised manner. For the selected views, we introduce `viewNMS` to remove redundant ones and improve the view *diversity*.

### 4.1. `viewAnnotator`

The implementation of `viewAnnotator` has two steps: caption generation and view matching, as shown in Figure 5. Both steps use the same LVLM as in the zero-shot 3D-QA (*i.e.*, the final inference model). This process aims to identify the critical views that match mostly the content of both input questions and the corresponding answers. Please note that these data are all from the training set where the answers are available for use.

**Caption Generation**. It begins by feeding a question-answer pair $(Q, A)$ and a rephrasing prompt ($Prompt_R$) into the LVLM, as in Step 1 of Figure 5. This prompt is fixed for every question-answer pair and instructs the model to rephrase the pair into an image caption $C$ which abridges the reasoning between the question and answer:

$$C = \text{LVLM}(Q, A, Prompt_R). \qquad (5)$$

Please note that caption generation is a crucial prior step of view matching. Directly using the $(Q, A)$ pair for matching causes the model to focus on answering the question rather than labeling the views. In other words, it encourages the model to take a shortcut by simply copying the answer $A$.

**View Matching**. For each view $V_i$ in a set of 2D views

$\mathcal{M}$, we evaluate its information relevance to the generated caption using LVLM. Specifically, we prompt the caption $C$ and a matching prompt $Prompt_M$, as in Step 2 of Figure 5, to LVLM. LVLM classifies $V_i$ into one of three categories, "positive", "negative", or "uncertain", respectively corresponding to the options A, B, and C in $Prompt_M$.

$$S_i = \text{LVLM}(C, V_i, Prompt_M), \qquad (6)$$

where $S_i \in \{0, 1\}$ is the classification label of the view $V_i$. For example, in Figure 5, a view is classified as "positive" ($S_i = 1$) because it contains the correct objects with specified attributes and spatial relationships, such as a "brown cabinet" with a "television" inside and "curtains" nearby. Otherwise, views are labeled as "negative" ($S_i = 0$). Views are classified as "uncertain" when the model chooses the option of "Uncertain, insufficient or unclear information" or outputs none of the given options, and these views are excluded from training.

## 4.2. `viewSelector`

As shown in Figure 3, `viewSelector` is plugged between the visual encoder and LVLM to select "views" in the feature space. It takes the question embedding $\mathbf{Q}$ and the visual embedding set $\{\mathbf{V_i}\}_{i=1}^N$ as input and outputs a binary label $\hat{S}_i$ (0 or 1) for each visual embedding. Then, $\hat{S}_i$ is compared to the corresponding view label generated by the `viewAnnotator`. The mismatch loss is used to optimize the parameters of `viewSelector`.

Specifically, the question embedding $\mathbf{Q}$ is first passed through a linear layer. followed by a two-layer Transformer block, and a pooling layer. The output can be regarded as a compact summary of the question, producing a question vector $\mathbf{q}$. Similarly, for visual inputs, each visual embedding $\mathbf{V}_i$ is processed through the same modules. We apply cross-attention in each transformer layer between the question embedding $\mathbf{Q}$ and the visual embeddings $\{\mathbf{V}_i\}_{i=1}^N$, in order to enhance the model's ability to identify views containing critical content for QA. After pooling, the resulting set of vectors $\{\mathbf{v}_i\}_{i=1}^N$ serve as compact summaries of question-aligned visual embeddings.

Finally, the outputs $\mathbf{q}$ and $\{\mathbf{v}_i\}_{i=1}^N$ are used to measure the criticality between the question and each view by cosine similarity:

$$\hat{S}_i = \frac{\mathbf{q} \cdot \mathbf{v}_i}{||\mathbf{q}|| ||\mathbf{v}_i||}. \qquad (7)$$

The score $\hat{S}_i$ is supervised with the corresponding label $S_i$ by binary cross-entropy loss:

$$\mathcal{L}_{\text{BCE}} = -\frac{1}{N'} \sum_{i=1}^{N'} \left( \hat{S}_i \log(S_i) + (1 - \hat{S}_i) \log(1 - S_i) \right) \qquad (8)$$

where $N' \leq N$ is the number of views labeled as 1 ("positive") or 0 ("negative").

During inference, `viewSelector` acts as a scoring function to evaluate each input view: a higher score $\hat{S}_i$ indicates higher criticality of $V_i$.

## 4.3. `viewNMS`

The views selected by `viewSelector` may introduce redundancy: overlapping views might all get high scores—similar to the problem of image-retrieval-based methods. We propose `viewNMS` to filter out redundant views. We leverage camera parameters, *i.e.*, position and orientation, calculate distances between selected views, and discard views less distant than a predefined distance threshold.

Specifically, `viewNMS` operates in three steps: 1) **Ranking views** sorts all views $\{V_i\}_{i=1}^N$ by their scores $\{\hat{S}_i\}_{i=1}^N$ in descending order, resulting in $\{V_{i_k}\}_{k=1}^N$, where $I_{i_1}$ is the highest-scoring view. 2) **Initializing candidate views** selects the highest-scoring view as the initial set $\mathcal{M}' = \{V_{i_1}\}$. 3) **Adding diverse views** sequentially processes the remaining views in sorted order, adding a view $V_{i_k}$ to the set if its distance from previously selected views exceeds a threshold $T$, formulated as:

$$\mathcal{M}' = V_{i_k} \cup \mathcal{M}', \text{if } D(V_{i_k}, V_j) > T, \forall V_j \in \mathcal{M}'. \qquad (9)$$

Finally, `viewNMS` outputs a new set of selected views $\mathcal{M}'$, which are both critical and spatially diverse. After that, $\mathcal{M}'$ and $Q$ are fed into the 2D LVLM to generate an answer which is the final output of zero-shot 3D-QA.

**View Distance Calculation**. The core of `viewNMS` lies in the calculation of the view distance, *i.e.*, $D(V_i, V_j)$, measuring the cameras' position and orientation distance between $V_i$ and $V_j$. For each view, the camera parameters $[\mathbf{R}|\mathbf{t}]$ (we omit the subscript for simplicity) consist of a camera orientation $\mathbf{R} \in \mathbb{R}^{3 \times 3}$ and a camera position $\mathbf{t} \in \mathbb{R}^{3 \times 1}$. The distance is calculated by combining both the orientation distance and position distance. For the orientation $\mathbf{R}$, we first convert it into a quaternion representation $\mathbf{p} = [p_x, p_y, p_z, p_w]$ for more efficient distance calculations. Then, the orientation distance $D_{ori}(V_i, V_j)$ is calculated by

$$D_{ori}(V_i, V_j) = 2 \cdot \arccos(|\mathbf{p}_i \cdot \mathbf{p}_j|), \qquad (10)$$

where $\arccos$ represents the inverse cosine function. This formula gives the angular distance in radians between the orientations of two views. Since $\arccos(|\mathbf{p}_i \cdot \mathbf{p}_j|)$ yields half the angle, the factor of 2 restores the full angle difference.

The position distance $D_{pos}(V_i, V_j)$ between views $V_i$ and $V_j$ is calculated using the Euclidean distance between their camera positions $\mathbf{t}_i$ and $\mathbf{t}_j$:

$$D_{pos}(V_i, V_j) = ||\mathbf{t}_i - \mathbf{t}_j||, \qquad (11)$$

| Method | Type | ScanQA | | | | SQA |
| | | EM@1 | BLEU-1 | ROUGE | CIDEr | EM@1 |
|---|---|---|---|---|---|---|
| ScanQA (Azuma et al., 2022) | 3D | 23.5 / 20.9 | 31.6 / 30.7 | 34.3 / 31.1 | 67.3 / 60.2 | 45.3 |
| SQA3D (Ma et al., 2022) | 3D | - | - | - | - | 47.2 |
| 3D-LLM (Hong et al., 2023) | 3D | 19.1 / - | 38.3 / - | 35.3 / - | 69.6 / - | 48.1 |
| 3D-VLP (Jin et al., 2023a) | 3D | 24.6 / 21.6 | 33.2 / 31.5 | 36.0 / 31.8 | 70.2 / 63.4 | - |
| 3D-VisTA (Zhu et al., 2023) | 3D | 27.0 / 23.0 | - | 38.6 / 32.8 | 76.6 / 62.6 | 48.5 |
| SIG3D (Man et al., 2024a) | 3D | - | - | - | - | 52.6 |
| SynFormer3D (Yang et al., 2024) | 3D | 27.6 / 24.1 | - | 39.2 / 33.3 | 76.2 / 62.7 | - |
| LL3DA (Chen et al., 2024a) | 3D+2D | - | - | 38.2 / 35.2 | 78.2 / 70.3 | - |
| PQ3D (Zhu et al., 2025) | 3D+2D | 26.1 / 20.0 | 43.0 / 36.1 | - | 87.8 / 65.2 | 47.1 |
| BridgeQA (Mo & Liu, 2024) | 3D+2D | 31.3 / 30.8 | 34.5 / 34.4 | 43.3 / 41.2 | 83.8 / 79.3 | 52.9 |
| LLAVA-OV + $\mathcal{F}_{\text{uniform}}$ | 2D | 33.1 / 33.5 | 43.2 / 44.2 | 46.9 / 46.6 | 95.8 / 93.3 | 53.5 |
| LLAVA-OV + $\mathcal{F}_{\text{retrieval}}$ | 2D | 33.9 / 34.6 | 44.8 / 46.1 | 48.3 / 48.7 | 98.8 / 97.7 | 55.0 |
| LLAVA-OV + $\mathcal{F}_{\text{cdViews}}$ | 2D | **35.0 / 35.6** | **46.1 / 47.2** | **49.7 / 49.5** | **102.8 / 100.4** | **56.9** |
| *margin over the compared best* | - | 3.7 ↑ / 4.8 ↑ | 3.1 ↑ / 9.1 ↑ | 6.4 ↑ / 8.3 ↑ | 15.0 ↑ / 21.1 ↑ | 3.9 ↑ |

Table 1: Performance comparisons with the state-of-the-art methods on the test set of ScanQA (Azuma et al., 2022) and SQA (Ma et al., 2022). For ScanQA, scores are presented in the format "with object test set" / "without object test set". The best and second best results are in **bold** and underlined, and the last row shows the performance margins between LLAVA-OV + $\mathcal{F}_{\text{cdViews}}$ and the top-performing related methods.

where $|| \cdot ||$ is the Euclidean norm.

The final camera distance $D(V_i, V_j)$ is a sum of the position and orientation distances,

$$D(V_i, V_j) = D_{pos}(V_i, V_j) + D_{ori}(V_i, V_j). \quad (12)$$

Combining the camera's position and orientation, the distance estimates the spatial overlap between the regions captured by two views, with smaller values indicating greater overlap. An ablation study on threshold selection is provided in the experimental section.

## 5. Experiments

**Datasets**. We use ScanQA (Azuma et al., 2022) and SQA (Ma et al., 2022) in our experiments, both constructed from ScanNet dataset (Dai et al., 2017). ScanQA contains over 41K question-answer annotations across 800 indoor 3D scenes, which are divided into train, val, and test sets (with or without objects). SQA contains over 33K question-answer pairs derived from 650 indoor scenes. It encompasses a diverse range of question types, including object identification, spatial relationships, scene-level understanding, and general reasoning.

**Evaluation Metrics**. We adopt Exact Match (EM@1) for both datasets. EM@1 measures the proportion of cases where the top-1 predicted answers match any of the ground-truth answers. Furthermore, since the answers in ScanQA are often free-form, we use standard text similarity metrics, including BLEU-1 (Papineni et al., 2002), ROUGE-L (Lin, 2004), and CIDEr (Vedantam et al., 2015) to assess the quality of generated answers.

**Implementation Details**. We utilize a recent state-of-the-art LVLM, *i.e.*, LLAVA-OV-7B (Li et al., 2024a), as the 2D LVLM for all experiments, including viewAnnotator and 3D-QA. The model remains frozen throughout all experiments. Analysis on more LVLM backbones is shown in Appendix C. The only trainable component is viewSelector, a lightweight module with a total of $5.9M$ parameters. Training of the viewSelector is conducted with a learning rate of $5 \times 10^{-5}$ and a batch size of 8. Each training iteration samples 5 positive and 5 negative views per instance generated by viewAnnotator. Here the number of views, *e.g.*, $k = 9$ for cdViews, is selected on the validation set (Figure 4).

### 5.1. Comparisons with the State-of-the-Arts

Table 1 presents the quantitative results comparing 2D-only methods (uniform sampling, image retrieval, and cdViews) with other LLAVA-OV (Li et al., 2024a) with state-of-the-art 3D and hybrid methods. First, it is observed that 2D-only methods achieve superior performance, showing the advantage of applying 2D pre-trained models for 3D tasks. For example, compared to BridgeQA (Mo & Liu, 2024), our $\mathcal{F}_{\text{cdViews}}$ achieves significant improvements of 15.0% and 21.1% CIDEr on the two test sets of ScanQA. Second, among the 2D-only methods, cdViews outperforms the others. For example, $\mathcal{F}_{\text{cdViews}}$ outperforms $\mathcal{F}_{\text{retrieval}}$ by 4.0% and 2.7% CIDEr on both test sets of ScanQA. The reason is that the uniform sampling method ignores the question and the image retrieval method often fails to capture critical views or introduces redundancy views. In contrast, cdViews effectively identifies critical and diverse views for efficient 3D-QA. The qualitative comparison of

<Question >: What is to the right of the chair?                                                    <Answer >: desk
▲BridgeQA: ottoman   | ★ LLAVA-OV + $\mathcal{F}_{retrieval}$: couch   |● LLAVA-OV + $\mathcal{F}_{cdView}$ : desk

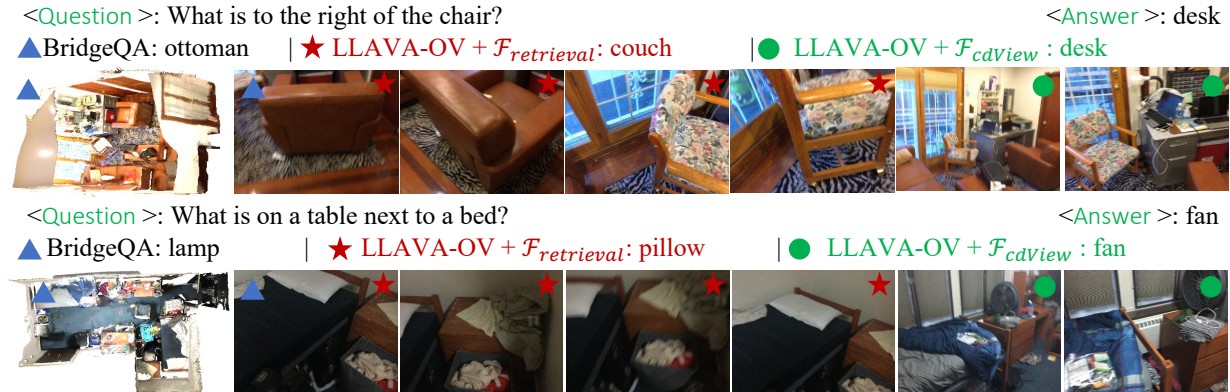

<Question >: What is on a table next to a bed?                                                     <Answer >: fan
▲ BridgeQA: lamp   | ★ LLAVA-OV + $\mathcal{F}_{retrieval}$: pillow   |● LLAVA-OV + $\mathcal{F}_{cdView}$ : fan

Figure 6: **Qualitative results** for BridgeQA (Mo & Liu, 2024), LLAVA-OV + $\mathcal{F}_{retrieval}$, and our final model LLAVA-OV + $\mathcal{F}_{cdViews}$. The marks ▲, ★, and ● represents the selected views respectively by three methods. We can see that `cdViews` captures the most critical and diverse views to answer the questions.

| LLAVA-OV | view Selector | view NMS | Best EM@1 | Optimal $k$ |
|---|---|---|---|---|
| + $\mathcal{F}_{\text{uniform}}$ | - | - | 28.3 | 17 |
| + $\mathcal{F}_{retrieval}$ | - | - | 29.1 | 17 |
| + $\mathcal{F}_{retrieval}$ | - | ✓ | 29.2 | 9 |
| + $\mathcal{F}_{cdViews}$ | ✓ | - | 29.7 | 17 |
| + $\mathcal{F}_{cdViews}$ | ✓ | ✓ | 30.1 | 9 |

Table 2: An ablation study performed on ScanQA. We show the best EM@1 scores with the corresponding (optimal) $k$.

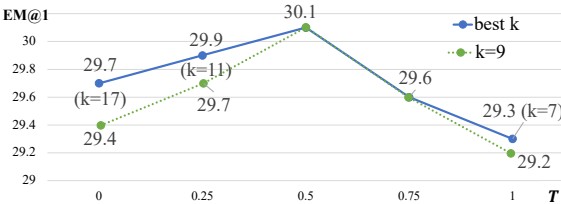

Figure 7: The results of EM@1 using two configurations: optimal $k$ (blue) *vs.* fixed $k=9$ (green). X-axis is the threshold $T$ of `viewNMS`. $T=0$ means disabling `viewNMS`.

selected views is shown in Figure 2 and Figure 6. More comparisons are provided in the Appendix Section B.

**5.2. Ablation Study**

In this section, we conduct an ablation study on the validation set of ScanQA (Azuma et al., 2022), following (Mo & Liu, 2024). We study the impact of `cdViews` components and `viewNMS` thresholds. In addition, we particularly compare ours with the most related work: image-retrieval-based 3D-QA (Mo & Liu, 2024). More ablation studies are in Section C of the Appendix.

**cdViews Components.** The experimental results are summarized in Table 2. The first row shows the baseline performance using randomly sampled 2D views as input, *i.e.* $\mathcal{F}_{\text{uniform}}$, achieving the best result of 28.3% EM@1 with 17 views. The second and third rows present results using the image retrieval baseline. Compared to uniform sampling, retrieval provides better views and improves EM@1 to 29.1% (with 17 views). When combined with `viewNMS`, the number of input views is reduced to 9, and performance slightly improves to 29.2%. The fourth row presents the performance of $\mathcal{F}_{cdViews}$ with the `viewSelector` alone, which achieves 29.7% EM@1 with 17 views, improving by 1.4%. This validates that the `viewSelector` effectively prioritizes critical views. The last row reports the full

implementation of $\mathcal{F}_{cdViews}$, where `viewNMS` reduces the input to just 9 views—almost half the visual token length—without reducing the performance, but further boosting EM@1 by 0.4%. This is due to the reduced redundancy allowing the model to focus more on critical views. A comparison between the third and last rows shows that our full pipeline $\mathcal{F}_{cdViews}$ outperforms the retrieval + `viewNMS` baseline by 0.9% EM@1 (30.1% vs. 29.2%), using the same number of input views. Even after redundancy removal via `viewNMS`, the retrieval-based approach remains constrained by its initial candidate views, which are selected based on question–view semantic similarity rather than their criticality to question answering. This further highlights the strength of our learned `viewSelector`, which explicitly identifies views that are critical for question answering.

**viewNMS Thresholds**. We evaluate the effect of different `viewNMS` thresholds (0, 0.25, 0.5, 0.75, and 1.0) in Figure 7. As the threshold increases, the optimal number of input views decreases from 17 to 9, demonstrating the effectiveness of `viewNMS` in reducing redundancy. The highest accuracy is achieved at a threshold of 0.5, with only 9 views input. When the number of views is fixed at 9, performance improves with increasing thresholds, peaking at 0.5 before declining. It indicates that excessively high thresholds may

| Method | $\mathcal{F}_{\text{retrieval}}$ | $\mathcal{F}_{\text{cdViews}}$ |
|---|---|---|
| Model | BLIP$_{\text{ViT-L}}$ (retrieval) | cdViews |
| Parameters | 644M | 5.9M ($-99.1\%$) |
| FLOPs | 593.6T | 294.5T ($-50.4\%$) |
| Inference Time | 2.8s | 1.2s ($-57.1\%$) |

Table 3: Computational performance comparison between image retrieval and cdViews for zero-shot 3D-QA.

loss spatially close views, and thus miss critical information.

**cdViews's Efficiency**. We compare the efficiency of image retrieval and our proposed cdViews in Table 3. As a lightweight plug-in module to LVLMs, $\mathcal{F}_{\text{cdViews}}$ only introduces $5.9M$ parameters, while the parameters of $\mathcal{F}_{\text{retrieval}}$ is 100 times as $\mathcal{F}_{\text{cdViews}}$. Furthermore, $\mathcal{F}_{\text{cdViews}}$ reduces FLOPs by half and cuts inference time by more than 50% compared to $\mathcal{F}_{\text{retrieval}}$. These results demonstrate the effectiveness of cdViews in improving accuracy, streamlining inference, and reducing computation.

## 6. Conclusions

In this work, we leverage 2D LVLMs in a zero-shot manner (or plugging a lightweight module) to address 3D-QA and identify view selection as a critical factor affecting performance. Our preliminary study reveals that effective view selection must ensure both critical and diversity. To this end, we propose cdViews, a view selection framework comprising viewSelector, which prioritizes critical views, and viewNMS, which enhances spatial diversity by removing redundant views. Extensive experiments on the ScanQA and SQA datasets demonstrate that cdViews achieves state-of-the-art performance.

## Acknowledgments

This research is supported by the RIE2025 Industry Alignment Fund – Industry Collaboration Projects (IAF-ICP) (Award I2301E0026), administered by A*STAR, as well as supported by Alibaba Group and NTU Singapore, and the Major Research Program of Jiangsu Province (Grant BG2024042).

## Impact Statement

This paper presents work aiming to advance machine learning by introducing cdViews. It integrates a viewSelector and viewNMS to automatically select critical and diverse views for 3D question answering (3D-QA). By relying solely on 2D views and pre-trained LVLMs, this approach addresses the challenge of limited 3D training data and avoids the need for direct alignment between 3D and language representations. The proposed method demonstrates state-of-the-art performance on benchmarks such as ScanQA and SQA,

showing the potential of 2D LVLMs as effective alternatives to resource-intensive 3D LVLMs. Potential societal consequences include improvements in autonomous systems, assistive technologies, and interactive environments, where efficient 3D scene understanding is critical. While no immediate risks or concerns are identified.

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

This supplementary includes the details of view matching in `viewAnnotation` (Sec. A), more comparisons with the State-of-the-Arts (Section B), more ablation studies (Section C), including ablation of view selection methods with different 2D LVLM, effectiveness of caption generation in `viewAnnotator`, and more case studies (Section D).

## A. More Details in View Matching

**This supplementary is for Sec. 4.1 of the main paper**. In our `viewAnnotator`, view matching classifies views as positive, negative, or uncertain. However, directly using a 2D LVLM with the prompt $Prompt_M$ as an instruction is unreliable, as the model lacks an explicit judgment criterion. To address this, we leverage its strong in-context learning ability (Zhou et al., 2024) by providing a textual context example that guides the model through a structured reasoning process. Specifically, we incorporate a step-by-step system prompt in the View Matching process. As shown in Figure S1, the system prompt ensures that all key objects, attributes, and spatial relationships in the caption align with the image, reducing ambiguity and improving consistency. Uncertain views are explicitly excluded, enhancing the robustness of the annotation process. Additional examples of positive and negative views are shown in Figure S2.

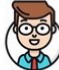

*\<System Prompt\>*: Consider the following example to guide your responses:
Caption: "A brown cabinet with a television inside is located in the right corner of the room, near the curtains. "
In this example, following the steps:
1. List all objects or elements mentioned in the caption:
  - Brown cabinet   - Television inside the cabinet   - Curtains nearby
2. Check if all objects from the caption are present in the image:
  - Yes, if all objects from the caption (brown cabinet, television, and curtains) are present in the image, proceed to step 3.
  - No, answer with option B.
3. Verify if the objects' attributes and relative positions match the caption:
  - Yes, the cabinet is brown, the television is inside the cabinet, it is positioned in the right corner, and it is near the curtains.
  - If any attributes or positions do not match the caption, answer with option B.
  - If the image contains partial but unclear information, answer with option C.

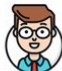

*\<$Prompt_M$\>*: You are given an image and a caption describing the visual content. Determine if the image matches the caption, and respond with one of the following options:
A. Yes, fully matches.    B. No, does not match.    C. Uncertain, insufficient or unclear information.

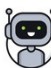

*\<Caption\>*:an orange storage bin is placed on top of a white cabinet.

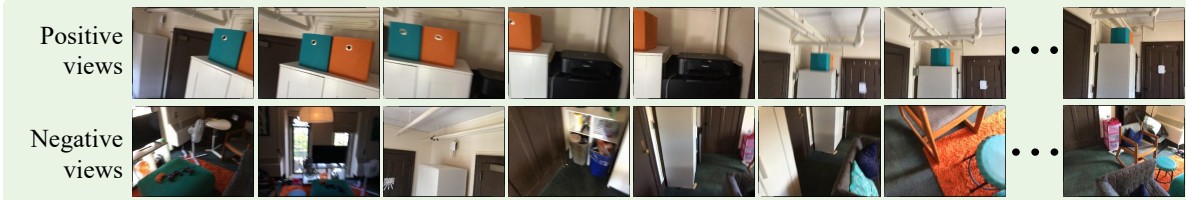

Figure S1: **Illustration of how context guides the view matching process**. In the view matching process of `viewAnnotator`, the model follows a structured reasoning approach, using a textual example to classify views as positive, negative, or uncertain.

To further validate the reliability of the positive views, we conducted a human evaluation: We randomly selected 50 QA pairs with their associated positive views. Three human evaluators assessed whether each view could answer the question. Their accuracy rates were 96.72%, 94.28%, and 97.56%, confirming that the quality of positive views is sufficient for training.

## B. More Comparisons with the State-of-the-Art Methods

**This supplementary is for Section 5.1 of the main paper**. Table S1 presents the quantitative results comparing LLAVA-OV (Li et al., 2024a) with different view selection methods, including uniform sampling, image retrieval, and our `cdViews`, against state-of-the-art methods on the validation set of ScanQA. As shown, LLAVA-OV + $\mathcal{F}_{\texttt{cdViews}}$ outperforms these

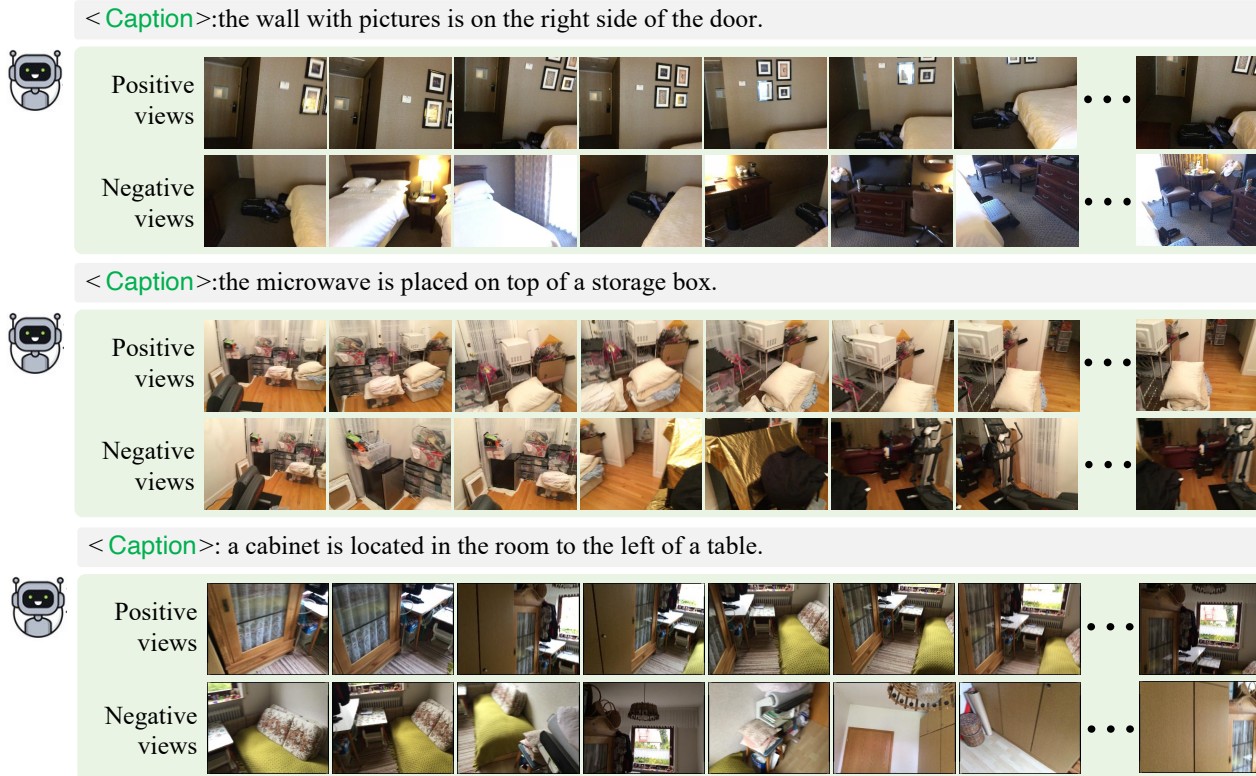

Figure S2: **Examples of automatically annotated positive and negative views**. Each case shows a caption along with its corresponding positive and negative views. Positive views closely match the caption in terms of key objects, attributes, and spatial relations, while negative views lack full correspondence.

| Method | Type | EM@1 | BLEU-1 | ROUGE | CIDEr |
|---|---|---|---|---|---|
| ScanQA (Azuma et al., 2022) | 3D | 20.3 | 29.5 | 32.4 | 61.7 |
| 3D-LLM (Hong et al., 2023) | 3D | 20.5 | 39.3 | 35.7 | 69.4 |
| 3D-VLP (Jin et al., 2023a) | 3D | 21.7 | 30.5 | 34.5 | 67.0 |
| LL3DA (Chen et al., 2024a) | 3D+2D | - | - | 37.3 | 76.8 |
| BridgeQA (Mo & Liu, 2024) | 3D+2D | 27.0 | - | - | - |
| GPT-4O+CC (Liu et al., 2024a) | 2D | - | 35.4 | 42.6 | 87.0 |
| LLAVA-OV + $\mathcal{F}_{uniform}$ | 2D | 28.3 | 40.2 | 44.5 | 88.0 |
| LLAVA-OV + $\mathcal{F}_{retrieval}$ | 2D | 29.1 | 41.5 | 45.8 | 91.6 |
| LLAVA-OV + $\mathcal{F}_{\texttt{cdViews}}$ | 2D | **30.1** | **42.6** | **46.8** | **94.0** |
| *margin over the compared best* | | 3.1 ↑ | 3.3 ↑ | 4.2 ↑ | 7.0 ↑ |

Table S1: Result comparisons with the state-of-the-art methods on the validation set of ScanQA (Azuma et al., 2022). The best and second best results are in **bold** and underlined.

methods by clear margins. The last row of Table 1 highlights the performance gap between LLAVA-OV + $\mathcal{F}_{\texttt{cdViews}}$ and the best-performing baselines. Even compared to GPT-4O+CC (Liu et al., 2024a), which leverages the powerful capabilities of GPT-4O (OpenAI, 2024), LLAVA-OV + $\mathcal{F}_{\texttt{cdViews}}$ surpasses it by 7.0% CIDEr. GPT-4O+CC improves spatial understanding by adding object markers to track correspondences across uniformly sampled views. However, it overlooks the relevance between the selected views and the input question, limiting its effectiveness in 3D-QA.

Table S2 presents the quantitative results on SQA (Ma et al., 2022), detailing performance across different question types: "What", "Is", "How", "Can", "Which", and "Other". Compared to state-of-the-art methods, LLAVA-OV + $\mathcal{F}_{\texttt{cdViews}}$ achieves

the best performance on "What", "How", "Which", and "Other" questions but shows a decline of 2.8% and 10.2% on "Is" and "Can" questions, respectively. This decline may be attributed to the zero-shot nature of LLAVA-OV (Li et al., 2024a), which maintains balanced performance across all question types. In contrast, other methods exhibit uneven performance, excelling in Is" and Can" questions due to dataset-specific adaptation while potentially underperforming in other categories. Furthermore, based on the same 2D LVLM, LLAVA-OV, cdViews consistently outperforms uniform sampling and image retrieval across all question types, demonstrating its effectiveness in selecting critical views for 3D-QA.

| Method | Input | Question Breakdown | | | | | | Overall |
|---|---|---|---|---|---|---|---|---|
| | | What | Is | How | Can | Which | Other | |
| GPT-3 (Brown et al., 2020) | 3D | 39.7 | 46.0 | 40.5 | 45.6 | 36.1 | 38.4 | 41.0 |
| ScanQA (Azuma et al., 2022) | 3D | 28.6 | 65.0 | 47.3 | 66.3 | 43.9 | 42.9 | 45.3 |
| SQA3D (Ma et al., 2022) | 3D | 33.5 | 66.1 | 42.4 | 69.5 | 43.0 | 46.4 | 47.2 |
| 3D-LLM (Hong et al., 2023) | 3D | 36.5 | 65.6 | 47.2 | 68.8 | 48.0 | 46.3 | 48.1 |
| 3D-VisTA (Zhu et al., 2023) | 3D | 34.8 | 63.3 | 45.4 | 69.8 | 47.2 | 48.1 | 48.5 |
| SIG3D (Man et al., 2024a) | 3D | 35.6 | **67.2** | 48.5 | **71.4** | 49.1 | 45.8 | 52.6 |
| PQ3D (Zhu et al., 2025) | 3D+2D | 37.1 | 61.3 | 44.5 | 60.9 | 47.0 | 45.1 | 47.1 |
| BridgeQA (Mo & Liu, 2024) | 3D+2D | - | - | - | - | - | - | 52.9 |
| LLAVA-OV + $\mathcal{F}_{\text{uniform}}$ | 2D | 51.4 | 60.7 | 49.6 | 56.2 | **51.6** | 51.9 | 53.5 |
| LLAVA-OV + $\mathcal{F}_{\text{retrieval}}$ | 2D | 54.8 | 62.4 | 50.3 | 56.5 | 49.3 | 53.2 | 55.0 |
| LLAVA-OV + $\mathcal{F}_{\text{cdViews}}$ | 2D | **55.0** | 64.4 | **54.0** | 61.2 | **51.6** | **54.4** | **56.8** |
| *margin over the compared best* | | 15.3 ↑ | −2.8 ↓ | 5.5 ↑ | −10.2 ↓ | 2.5 ↑ | 6.3 ↑ | 3.9 ↑ |

Table S2: Result comparisons with the state-of-the-art methods on the test set of the SQA (Ma et al., 2022). The best and second-best results are in **bold** and underlined. The decline in the "Is" and "Can" problems for LLAVA-OV with different view selections is attributed to the zero-shot nature of LLAVA-OV, which ensures balanced performance across all question types. In contrast, the compared methods exhibit uneven performance, excelling in "Is" and "Can" questions due to dataset-specific adaptation while potentially underperforming in other categories.

## C. More Ablation Studies

**This supplementary is for Section 5.2 of the main paper**. The ablation studies are conducted on the validation set of ScanQA (Azuma et al., 2022), we evaluate the impact of different backbones, the effect of caption generation in viewAnnotator, and visualize the views within different distance thresholds, and visually compare the results of BridgeQA, LLAVA-OV + $\mathcal{F}_{\text{retrieval}}$, and LLAVA-OV + $\mathcal{F}_{\text{cdViews}}$.

**Ablation with Different Backbones**. We evaluate the impact of different backbones by comparing LLAVA-NEXT (Li et al., 2024b) and LLAVA-OV (Li et al., 2024a), with results presented in Table S3. The results reveal two key insights: 1) View selection plays a crucial role in enhancing performance across models. Replacing uniform sampling with image retrieval improves performance by 1.1% on LLAVA-NEXT and 0.8% on LLAVA-OV, underscoring the importance of selecting informative views for 3D-QA. Our cdViews further amplifies these gains, achieving improvements of 3.6% and 1.8%, respectively, by effectively identifying more critical views. 2) cdViews demonstrates robustness and adaptability, consistently outperforming both baselines and delivering the highest performance gains across all evaluation metrics.

**Effectiveness of Caption Generation in `viewAnnotator`**. To assess the necessity of caption generation for view matching, we conduct an ablation study by removing the caption generation step in viewAnnotator. Instead of using the generated caption $C$, the question-answer pair $(Q, A)$ is directly used as input for view matching. To better isolate the impact of caption generation, this ablation study is conducted without applying viewNMS. Specifically, Eq. 6 is modified as:

$$\bar{S}_i = \text{LVLM}(Q, A, V_i, Prompt'_M), \tag{13}$$

where $Prompt'_M$ is an adapted version of $Prompt_M$, with the term "caption" replaced by "question-answer pair." The textual context example is preserved to guide the view labeling step-by-step. The results, presented in Table S4, show that removing the caption generation step leads to a 1.8% drop in CIDEr for LLAVA-OV + $\mathcal{F}_{\text{cdViews}}$. This highlights the

| Backbone | View Selection | EM@1 | BLEU-1 | ROUGE | CIDEr |
|---|---|---|---|---|---|
| LLAVA-Next | + $\mathcal{F}_{\text{uniform}}$ | 21.0 | 30.0 | 42.1 | 85.3 |
| | + $\mathcal{F}_{\text{retrieval}}$ | 22.1 $_{1.1\uparrow}$ | 35.2 $_{5.2\uparrow}$ | 42.3 $_{0.2\uparrow}$ | 87.0 $_{1.7\uparrow}$ |
| | + $\mathcal{F}_{\text{cdViews}}$ | 24.6 $_{3.6\uparrow}$ | 39.6 $_{9.6\uparrow}$ | 44.9 $_{2.8\uparrow}$ | 93.7 $_{8.4\uparrow}$ |
| LLAVA-OV | + $\mathcal{F}_{\text{uniform}}$ | 28.3 | 40.2 | 44.5 | 88.0 |
| | + $\mathcal{F}_{\text{retrieval}}$ | 29.1 $_{0.8\uparrow}$ | 41.5 $_{1.3\uparrow}$ | 45.8 $_{1.3\uparrow}$ | 91.6 $_{3.6\uparrow}$ |
| | + $\mathcal{F}_{\text{cdViews}}$ | **30.1** $_{1.8\uparrow}$ | **42.6** $_{2.4\uparrow}$ | **46.8** $_{2.3\uparrow}$ | **94.0** $_{6.0\uparrow}$ |

Table S3: Ablation study results with different backbone models, LLAVA-Next (Li et al., 2024b) and LLAVA-OV (Li et al., 2024a). The best results are in **bold**. Subscripts indicate the relative improvement over the corresponding baseline, *i.e.*, the 2D LVLM with uniform sampling.

| Method | View Matching with Input Tuple | EM@1 | BLEU-1 | ROUGE | CIDEr |
|---|---|---|---|---|---|
| LLAVA-OV + $\mathcal{F}_{\text{cdViews}}$ | $(Q, A, V_i, Prompt'_M)$ | 29.5 | 41.4 | 45.9 | 91.4 |
| | $(C, V_i, Prompt_M)$ | 29.7 | 42.2 | 46.4 | 93.2 |

Table S4: Ablation study on the necessity of caption generation for view matching. The key difference lies in whether the `viewSelector` is trained with view labels generated using the caption $C$ or the $(Q, A)$ pair as input.

importance of generating a reformulated caption, which helps the model more effectively identify critical views compared to directly using the $(Q, A)$ pair.

**Effect of Finetuning LLaVA-OV in a Hybrid Method**. To assess the feasibility and effectiveness of incorporating LLAVA-OV into a hybrid method, we implement a variant of BridgeQA—the strongest hybrid baseline in our main comparisons. Specifically, we retain the original BridgeQA architecture but replace its 2D vision-language module (BLIP (Li et al., 2022)) with LLAVA-OV. For a fair comparison, we also replace its top-1 image input with 9 views selected by our `cdViews` strategy, while keeping the full-scene point cloud features extracted by VoteNet (Qi et al., 2019).

During training, we adopt parameter-efficient tuning by updating only the last 2 of the 28 transformer layers in LLAVA-OV using LoRA (Hu et al., 2022). As shown in Table S5, the finetuned variant (BridgeQA$_{\text{LLAVA-OV}}$) achieves a +1.4% EM@1 improvement over the original BridgeQA baseline (28.4% vs. 27.0%), confirming the benefit of using a stronger 2D LVLM. Nonetheless, it still underperforms our `cdViews`, which achieves 30.1% EM@1. This experiment demonstrates that while hybrid pipelines can benefit from stronger LVLMs, they rely on complex architectures, 3D-specific modules, and computationally expensive fine-tuning. In contrast, our framework achieves superior performance by using a 2D-only LVLM in a zero-shot inference manner.

| Method | Input | 2D LVLM | EM@1 | BLEU-1 | ROUGE | CIDEr |
|---|---|---|---|---|---|---|
| BridgeQA | 3D+2D | BLIP | 27.0 | - | - | - |
| BridgeQA$_{LLAVA-OV}$ | 3D+2D | LLAVA-OV | 28.4 | 37.3 | 42.7 | 84.0 |
| cdViews | 2D | LLAVA-OV | 30.1 | 42.6 | 46.8 | 94.0 |

Table S5: Comparison between zero-shot `cdViews` and fine-tuned hybrid BridgeQA using LLaVA-OV. We compare the performance of the original BridgeQA, its fine-tuned variant with LLaVA-OV, and our zero-shot `cdViews`. While the hybrid variant benefits from a stronger LVLM, our approach outperforms it with a 2D-only LVLM in a zero-shot inference manner.

# D. More Case Studies

**Visualize Comparison of Different Methods and Their Visual Inputs**. Figure S3 presents a visual comparison of the predicted answers and visual inputs of BridgeQA (Mo & Liu, 2024), LLAVA-OV + $\mathcal{F}_{\text{retrieval}}$, and LLAVA-OV + $\mathcal{F}_{\text{cdViews}}$. BridgeQA relies on the top-1 image retrieval view combined with point clouds as input. However, relying on point clouds to

provide the whole scene often results in answers that miss critical details. For instance, in the $4_{th}$ row, while the model correctly mentions the trash can on the floor, it overlooks surrounding objects like the toilet, which is crucial for providing a more informative answer. For LLAVA-OV + $\mathcal{F}_{\text{retrieval}}$, image retrieval-based view selection may miss the critical views required for accurate answers. As shown in the $3_{rd}$ rows, the retrieved views tend to be redundant or incomplete. In the $3_{rd}$ row, the selected views focus on the cabinet beneath the window but omit the view displaying books on top, which is essential for correctly answering the question. In contrast, LLAVA-OV + $\mathcal{F}_{\text{cdViews}}$ selects critical and diverse views, capturing essential context and delivering accurate, informative answers.

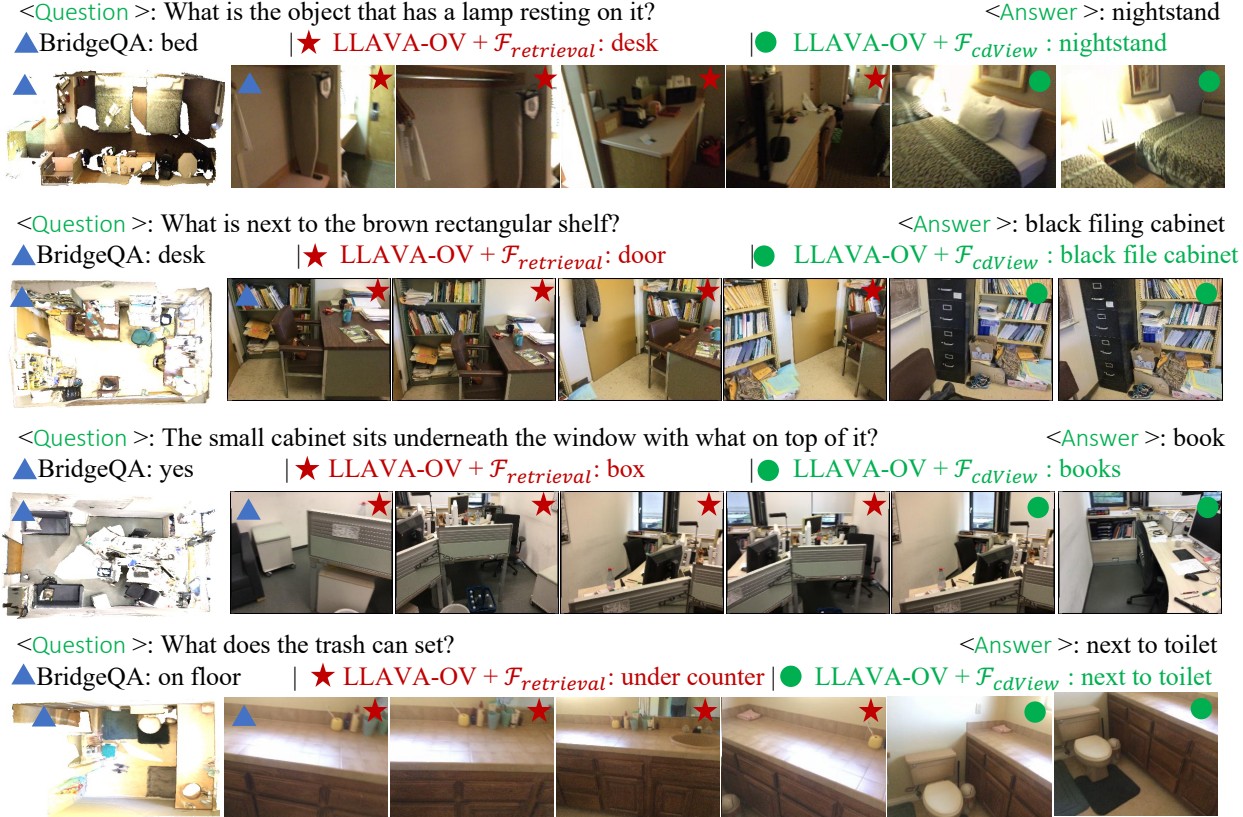

Figure S3: **More Qualitative results** for BridgeQA (Mo & Liu, 2024), LLAVA-OV + $\mathcal{F}_{\text{retrieval}}$, and our final model LLAVA-OV + $\mathcal{F}_{\text{cdViews}}$. Small marks ▲, ★, and ● represents the selected views by each method. We can see that cdViews can capture critical and diverse views to answer the questions.

