# OpenReview forum: "3D Question Answering via only 2D Vision-Language Models"
_ICML.cc/2025/Conference — ICML 2025 poster_

### Official Review · Reviewer_M1zQ · 2025-03-04

**Overall Recommendation:** 3

**Summary:**

This paper proposes to address the task of 3D question-answering (3D-QA). Proposed method takes as input a set of posed RGB images and operates by only using 2D large vision language models (in this case LLAVA-OV). The method does not operate on any 3D input, and instead uses the set of available images to answer questions in 3D-QA benchmarks. As the method only relies on images, the key challenge is to select informative and diverse views that correspond to the task and more helpful for answering the question being asked about the scene. To address this, the method proposes a view-selection module named cdViews, consisting of two sub-parts: (1) viewSelect is a module trained using annotations from 3D-QA datasets to identify most critical and relevant views for answering the question, and (2) viewNMS module essentially measures pairwise view overlaps to identify and select diverse views. The proposed method is evaluated on the ScanQA and SQA datasets.

### Update after rebuttal
I appreciate the authors’ rebuttal and the clarifications provided. After carefully reading the response and considering the other reviews and discussions, I agree that the main contribution of the paper lies in demonstrating that 2D VQA models can outperform 3D-based models on current 3D-QA benchmarks. This is indeed a noteworthy and interesting finding, suggesting that strong performance on these benchmarks is possible by only relying on large vision-language models (LVLMs). But this finding does not yet persuade me that LVLMs are necessarily stronger for the 3D-QA task in the general sense as claimed in this paper. This could even be instead pointing to the limitations of the existing 3D benchmarks in terms of evaluating spatial relations.

Overall, while the paper makes an interesting point, I believe the analysis of the results could be strengthened, especially in terms of exploring the role and limitations of 3D reasoning in the presented approach. As noted in my initial review, I had raised several questions related to this, such as whether LVLM-based methods are truly expected to model spatial relationships in 3D scenes. While the rebuttal addressed some of these concerns, I find that some of my reservations remain, particularly with respect to the 3D spatial reasoning ability of the method and the method limitations. In the light of this, I will be keeping my original score (3).

**Claims And Evidence:**

- The submission claims to address 3D-QA, yet the method relies solely on 2D images—with the only 3D aspect being the viewNMS module that uses camera poses to select diverse views if I understood correctly. This raises questions about whether the approach truly tackles 3D-QA, which typically requires a full 3D scene representation to capture spatial relationships. Moreover, since 2D images often suffice for answering questions about nearby object pairs, it remains unclear if the method robustly handles _spatial_ reasoning. I indeed find it impressive to see that this 2D-only method can perform much better than 3D-based methods, which is an interesting finding of this submission. But I am not yet fully convinced that these gains can be attributed to the method proposed in this work, for instance if one combines the baselines for 3D/2D hybrid methods with LLAVA-OV, would we see better results as they have access to 3D reconstruction as well (and can attend to the full 3D scene)? Also, further analysis of the types of questions answered well (and those not) by the proposed method could clarify whether the method truly understands broader spatial relations, such as identifying objects facing each other across a room and never visible together in a single image (e.g., "what is located right across the window" where the answer could be a TV far away).
- [L153-156] “We are the first to leverage 2D LVLMs in a zero-shot manner to address 3D understanding tasks.” - This claim is partially incorrect. Many prior works on open-vocabulary 3D scene understanding, such as OpenScene, OpenMask3D, and Open3DIS, already rely on 2D LVLMs to extract knowledge about a 3D scene in a zero-shot manner. While the claim might hold for this specific 3D-QA setting and the LVLMs such as LLAVA (instead of CLIP-style VLMs), it should be narrowed in scope to avoid misrepresentation.
- [L155-157]: "We identify view selection as a critical factor in this zero-shot setting, a challenge that has not been explicitly addressed." – Again, I think that the scope of this statement should be narrowed as there are many open-vocabulary 3D scene understanding methods where object-related view selection has been explored.
- [L125-127, right column]: "These methods improve 3D-QA performance, they come with trade-offs such as increased model complexity and data processing requirements. Furthermore, reconstructing 3D scenes from 2D views incurs significant computational costs, limiting the scalability in complex scenes." – While this argument is valid in terms of the costs associated with reconstruction, it does not consider the costs associated with running LVLMs multiple times on multiple views, which is the case in the proposed method.
- [L142-144]: "They (2D-based methods) care more about the evaluation itself rather than explicitly exploring new methods to improve 2D LVLMs performance in 3D-QA." – This statement is vague and should be better substantiated.

**Essential References Not Discussed:**

The paper provides a comprehensive discussion of related works. However, it should be more careful in making strong claims about being the first to leverage 2D LVLMs for 3D tasks ([L153-156]), as prior works in open-vocabulary 3D segmentation have done something similar. Additionally, the discussion of view selection could reference prior multi-view reasoning works.

**Experimental Designs Or Analyses:**

I think the experimental designs and analyses are generally sound, I did not identify a critical issue.

**Methods And Evaluation Criteria:**

Proposed method is reasonable, and the evaluation methodology follow the established benchmarks for the task of 3D-QA.

**Other Comments Or Suggestions:**

_Minor comments:_

- The problem formulation in [L162-164] defines the scene representation as a set of 2D views but does not explicitly include camera poses, which are necessary for the viewNMS module. This should be clarified.
- L323: “.. indicates more criticality of V_i.” should be “higher criticality”.
- L300 (right column): “omit the subscription for simplify” - “we omit the subscript for simplicity”
- L359 (right column): “5.1 Comparisons with the State-of-the-Arts” should be “State-of-the-Art”

**Other Strengths And Weaknesses:**

_Strengths:_
- It is a well-written paper with very clear explanations.
- The proposed methodology has an interesting take on the 3D-QA task, showing that even without having an explicit 3D scene representation it is possible to reason about the scene well compared to the purely 3D or hybrid 3D-2D methods for 3D-QA task. The findings are convincing, and I think the experimental analysis sheds light on the benefit of leveraging 2D LVLMs for 3D scene understanding tasks.
- I appreciated the comprehensive discussion on related works.
- The evaluation methodology is reasonable, and I also appreciated that the code is provided.

_Weaknesses:_
- I was unable to see a discussion on the limitations, which is especially important given that the proposed method does not make use of a 3D scene representation of the scene while answering the questions related to the 3D scene.
- As noted in the Claims and Evidence section, there are some open questions about the method’s 3D capabilities, especially its ability to capture relationships between objects that do not appear together in a single RGB frame. While this 2D-based approach shows promising performance, this might also reflect some limitations in the current evaluation benchmarks. In my opinion it is not substantially evaluated or demonstrated whether the method can capture broader-range spatial relationships.
- Given the method's reliance on LLAVA-OV, it is not entirely clear to me whether the cdView component substantially contributes to the overall performance, considering that even a uniform view sampling method outperforms several 3D-based baselines.
- Some claims such as the novelty of using 2D VLMs for 3D understanding in a zero-shot setting are a bit too strong and could benefit from  phrasing with more care.

**Questions For Authors:**

I am happy to reconsider my score based on the answers to the following questions and the remarks discussed earlier in the Claims section:

1. How does the proposed method fundamentally differ from video-based VQA approaches, given that it does not use an explicit 3D representation? Can it reliably reason about object relationships when two objects are never observed together in a single view?
2. How does the computational cost of running LLAVA-OV multiple times on multiple views compare to the cost of performing 3D-QA using existing 3D-based or 3D/2D-based methods? Is the reduction in computational cost by not having to reconstruct a 3D scene comparable to the difference between 3D-QA inference runtimes for the proposed method and 3D-based methods? Also, it appears to me that the proposed method needs to store all RGB images at all times if I understand correctly. How do the memory requirements compare?
3. The paper does not explicitly discuss its limitations as far as I could see. What are the main failure cases where the method struggles, particularly in complex or large-scale scenes where a broader-range 3D understanding might be necessary?
4. If LLAVA-OV were integrated into existing 3D/2D hybrid methods, would we expect it to outperform the proposed approach? Did the authors explore this possibility?

**Relation To Broader Scientific Literature:**

The paper provides a comprehensive discussion of related works, and I appreciated the systematic discussion with 3D, 2D as well as 3D/2D hybrid methods. However, the paper should be more careful in making strong claims about being the first to leverage 2D LVLMs for 3D tasks ([L153-156]) in a zero shot manner, as prior works in open-vocabulary 3D segmentation (OpenScene, OpenMask3D, Open3DIS etc.) have followed similar zero-shot approaches. This could also be a language specificity issue, as these open-vocabulary methods largely use LVLMs such as CLIP, whereas the proposed method focuses on methods like LLAVA with conversational abilities. I think this should be made more clear in the discussion in order to more accurately position the paper.

**Theoretical Claims:**

N/A

---

> ### Author Rebuttal · Authors · 2025-04-01
>
> ### **1. "... finetuning LLaVA-OV ..."**
> Due to space limitation, please kindly refer to the response of PgGq `2`
>
> ---
> ### **2. "... analysis of the types of questions" & "a discussion on the limitations ..."**
> We conduct a detailed analysis of question types on the SQA dataset, with per-type performance reported in Table S2 (Supplement). The results show that our zero-shot 2D LVLM-based method achieves strong performance across most question types, particularly excelling in “what”-, “which”-, and “other”-type questions.
>
> These categories primarily involve identifying objects and reasoning about their spatial configuration within the 3D scene. Our method, powered by multi-view aggregation, can aggregate multi-view observations to infer spatial layouts and object relations in 3D environments, even when key objects are never observed within a single view. As shown in [Figure R3](https://anonymous.4open.science/r/icml_rebuttal-35FF/failure_case_with_caption.pdf) (1st and 2nd sample), the model correctly infers relationships between spatially separated entities across the room.
>
> **Limitation**: In contrast, we observe relatively lower performance on “can”-type questions, which often involve agent-centric reasoning and require understanding how the scene would change with respect to the agent’s actions or viewpoint shifts (e.g., “Can I see the faucet if I turn around?”). These cases demand the modeling of dynamic viewpoints and agent-scene interaction, which are not fully captured by current 2D LVLMs in a frozen, zero-shot setting.
>
> Overall, our analysis confirms that frozen 2D LVLMs, when equipped with informed view selection, can already support a broad range of 3D spatial reasoning tasks.
>
> ---
> ### **3. "... broader spatial relations ..."**
> Our method is capable of reasoning about spatial relationships between objects that are never observed in the same view. As shown in the second example of [Figure R3](https://anonymous.4open.science/r/icml_rebuttal-35FF/failure_case_with_caption.pdf), the printer, trash can, and paper cutter are distributed across different views with no co-observability. Despite this, the model accurately infers their spatial arrangement and correctly answers the question.
>
> This demonstrates that, even without explicit 3D reconstruction, our approach can integrate multi-view information to support global spatial reasoning.
>
> ---
> ### **4. "Strong claims"**
> - [L153-156]: change from "3D understanding tasks" to "3D-QA tasks"
> - [L155-157]: correct to "... as a critical factor in zero-shot 3D-QA, for which there is a lack of an efficient solution in prior works."
> - [L125-127, right column]: correct to "... but rely on explicit 3D reconstruction, needing additional models and causing more processing steps. In contrast, our method uses 2D views and feeds them into a unified 2D LVLM, which makes a simpler pipeline".
> - [L142-144]: correct to "They focus more on evaluating pretrained 2D LVLMs on 3D-QA tasks, rather than developing approaches to adapt and improve their performance for spatial reasoning."
>
> ---
> ### **5. "... cdView component contributes to the overall performance ..."**
> We agree that LLaVA-OV provides a strong foundation, and even the uniform view sampling yields competitive results. This actually motivated us to explore more in the direction of leveraging well-trained 2D LVLMs for tackling 3D-QA tasks, and then we proposed the light-weight learnable module cdViews.
>
> As we can see in Table 1 (manuscript), using cdViews brings consistent and clear performance improvements over uniform sampling across multiple benchmarks. Specifically, it yields +2.0% / +2.1% EM@1 and +7.0% / +7.1% CIDEr gains on the ScanQA test set (w/o objects), and +3.4% EM@1 improvement on SQA. These gains can be regarded as "substantial", especially considering that prior work typically improve by only 1% EM@1 or 3–4% CIDEr (see Table 1).
>
> ---
> ### **6. "claims zero-shot ..."**
> Due to space limitation, please kindly refer to the response of rVhy `1`
>
> ---
> ### **7. “Problem formulation”**
> [Lines 162–164]: “..., each associated with a camera matrix containing the position and orientation.”
>
> ---
> ### **8. "L323, L300, L359"**
>  We thank the reviewer for pointing these typos out. We will fix them in the revised version.
>
> ---
> ### **9. "Differ from video-VQA"**
> Traditional video-based VQA methods, while also using 2D information, are designed to process sequential frames to understand temporal dynamics.  In contrast, cdViews focuses on selecting the most informative static 2D views from a 3D scene to answer questions about the 3D space.
>
> ---
> ### **10. "Computation cost and memory cost"**
> Due to space limitation, please kindly refer to the response of PgGq `3`

---

### Official Review · Reviewer_PgGq · 2025-03-12

**Overall Recommendation:** 2

**Summary:**

In this work the authors proposed to solve 3D question answering with 2D VLMs only. Specifically, 3D scenes are first rendered into 2D images, which are then used to prompt 2D VLMs (e.g., LLaVA-OneVision). Moreover the authors found that the key to good performance is how to select views that are most relevant to the question and answer. Compared to baseline view selectors, such as uniform sampling or image retrieval, training a view selector (along with a view NMS algorithm) to choose views that maximize the performance achieves the best performance.

## update after rebuttal

Following the rebuttal I raised some concerns regarding the contribution of this paper. Although I was looking forward to a meaningful discussion, reviewer pwsQ refused to defend his/her "4: Accept" decision or to engage in further discussion. I stand with reviewer M1zQ and recommend rejection.

**Claims And Evidence:**

The major finding of this paper is that 2D VLMs can achieve state-of-the-art 3D VQA performance, when compared with other 3D-LLMs. This argument is not convincing due to the unfair comparison adopted in the experiment section (see next question). Other smaller findings and arguments are mostly supported with clear evidence.

**Essential References Not Discussed:**

None.

**Experimental Designs Or Analyses:**

Again I have concerns regarding the experimental designs. See "Methods And Evaluation Criteria" part of my review.

**Methods And Evaluation Criteria:**

The proposed method and evaluation criteria is sound. However, the quantitative results may not be based on fair comparison. The VLM used in the proposed method, i.e., LLaVA-OV, is significantly better than all other 3D-LLMs, in terms of both the supervised fine-tuning (SFT) data, the vision encoder, and the language model (LLaMA-3 vs Vicuna v1). The LLaVA-OV naturally has much stronger visual understanding and reasoning abilities, which presents an unfair comparison with baseline methods. Experimental results in appendix also shows that a weaker VLM (LLaVA-NeXT) significantly hurts the performance.

**Other Comments Or Suggestions:**

My major concern is regarding the unfair comparison in the experimental results. This is mainly due to the fact that the authors are not retraining any of the 2D or 3D VLMs. For instance, finetuning LLaVA-OV following a strong 3D-VLM baseline could provide important insights to this problem. Most importantly the comparisons should be conducted with comparable vision encoders and large language models.

**Other Strengths And Weaknesses:**

In general this work discusses the choices of 2D and 3D VLM to achieve strong 3D spatial understanding and reasoning, given the limited 3D-related data and abundant 2D multi-modal data. This is an interesting and important question. However I have doubts about the experimental settings of this paper -- an unfair comparisons could produce very misleading results.

**Questions For Authors:**

1. How many input views are sampled, filtered, and fed into the large language model?
2. All frames, selected or not, go through the visual encoder. This may cause significant computational costs. What is the GFLOP of the proposed methods compared to baseline methods? What is the wall clock time of inference compared to baseline methods?
3. Also there should be ablation studies on the number of frames rendered and selected, e.g., the final performance v.s. the computational costs.

**Relation To Broader Scientific Literature:**

This work related to the broader discussion of spatial intelligence -- how to develop artificial intelligence with spatial understanding and reasoning, specifically the choice of the input modality and the design of the model (2D or 3D VLM). However, given the concerns regarding the experimental design, the main findings of this paper may not stand. The contribution of this paper to the broader research topic is limited.

**Theoretical Claims:**

No theoretical claims presented.

---

> ### Author Rebuttal · Authors · 2025-04-01
>
> ### **1. "... the unfair comparison ..."**
> We agree that LLaVA-OV is a strong 2D LVLM. However, we do not consider the comparison unfair, as our core motivation and contribution lie in exploring how to leverage powerful 2D LVLMs in a zero-shot manner for 3D-QA.
>
> - Compared to 3D-based methods, current 3D-LLMs are limited by the scarcity of large-scale 3D-language data. This **data bottleneck** is exactly what we aim to address. By contrast, our method directly utilizes pretrained 2D LVLMs, which benefit from tons of 2D vision-language data.
>
> - Compared to 2D-3D hybrid methods, it may seem fairer to replace the 2D module with LLaVA-OV. However, these hybrid pipelines typically require additional training to align 2D and 3D features. For large 2D LVLMs like LLaVA-OV, this alignment process incurs substantial computational cost. This introduces a significant **resource bottleneck**.
>
> - Our method is designed as **a practical solution under both data and resource constraints**, showing that strong off-the-shelf 2D LVLMs can be effectively used for 3D-QA. Moreover, the framework is model-agnostic and generalizable to future 2D LVLMs with improved capabilities.
>
> ---
> ### **2. "... finetuning LLaVA-OV ..."**
> The reviewer suggests fine-tuning LLaVA-OV within a 2D–3D hybrid pipeline, which could potentially benefit from both strong language–vision alignment and 3D spatial context. However, this setting introduces fundamental limitations: combining 2D LVLMs with 3D inputs requires additional model components, training supervision, and modality alignment, making the pipeline heavier and less scalable. As discussed in Lines 68–74 of our Introduction,
> "2D features extracted from LVLMs are already well-aligned with language, but further alignment with 3D features requires careful model design and advanced training techniques. "
>
> As requested, we implement a hybrid variant based on BridgeQA, the strongest hybrid baseline in our comparisons. Specifically, we follow the BridgeQA architecture and replace its 2D LVLM with LLaVA-OV. The input views are combined with question and 3D features. For fair comparsion, we use 9 views selected by cdViews as the input. Results are reported in Table R1.
>
> | Method                              | Type   | 2D LVLM   | EM@1  |
> |-------------------------------------|--------|-----------|-------|
> | BridgeQA           | 3D+2D  | BLIP      | 27.0  |
> | BridgeQA$_{LLAVA-OV}$               | 3D+2D  | LLAVA-OV  | 28.4  |
> | LLAVA-OV + $F_{cdViews}$    | 2D     | LLAVA-OV  | **30.1** |
>
> **Table R1**: *Evaluating fine-tuned LLaVA-OV in a 2D–3D hybrid pipeline.*
>
> ---
> ### **3. "... computation efficiency comparison ..."**
> - **Number of candidate views**: Each 3D scene contains between 6 and 382 RGB views, with an average of 79 views per question.
>
> - **Computation during view selection**: All candidate views are processed by the view selection module. However, this stage contributes only a small fraction of the total cost—less than 10% of overall FLOPs—as shown in Table R2.
>
> - **Comparison with image retrieval**:
> In the view selection stage, our method reduces FLOPs by 54.8% (26.5T vs. 58.6T) and runtime by 90% (0.04s vs. 0.40s).
> In the QA stage, FLOPs are reduced by 49.9% (268T vs. 535T), and inference time by 51.3% (1.16s vs. 2.38s).
>
> - **GPU memory usage**:
> As shown in Table R2, cdViews achieves a lower peak GPU memory cost, making it more suitable for deployment on devices with limited memory capacity.
>
> |     Method      |       **FLOPs**        |        |   **Inference Time**    |        |   **Peak GPU Memory Usage**    |        |
> |:---------------:|:----------------------:|:------:|:------------------------:|:------:|:------------------------:|:------:|
> |                 | View Selection         |  QA    | View Selection           |  QA    |View Selection           |  QA    |
> | $F_{uniform}$   | 0                      | 535T   | 0                        | 2.38s  |   0   |     26.96G |
> | $F_{retrieval}$ | 0.74T/view × 79 = 58.6T| 535T   | 0.40s                    | 2.38s  |   7.89G  |   26.96G |
> | $F_{cdViews}$   | 0.34T/view × 79 = 26.5T| 268T   | 0.04s                    | 1.16s  |   9.56G   | 21.54G  |
>
> **Table R2**: *Efficiency comparison between baseline method and our cdViews.*.
>
> ---
> ### **4. Ablation on the number of selected views and computational cost**
> We provide the requested ablation on computational cost with respect to the number of selected views. As shown below, FLOPs grow approximately linearly from 268 TFLOPs (9 views) to 534.98 TFLOPs (17 views). For the performance impact of view number, please kindly refer to Figure 4 of the main paper.
> | # Views | FLOPs (TFLOPs) |
> |--------:|----------------|
> |    9    | 268.00         |
> |   10    | 299.89         |
> |   11    | 332.19         |
> |   12    | 364.95         |
> |   13    | 398.12         |
> |   14    | 431.69         |
> |   15    | 465.71         |
> |   16    | 500.14         |
> |   17    | 534.98         |

---

### Official Review · Reviewer_pwsQ · 2025-03-13

**Overall Recommendation:** 4

**Summary:**

The paper aims to only use 2D vision-language models to address 3D question answering task. The authors propose a new framework cdViews, which select critical and diverse views and then perform 3D question answering using the 2D vision-language model. The proposed framework is evaluated on two widely used benchmarks and demonstrate the effectiveness.

## update after rebuttal
Please see the rebuttal comment below.

**Claims And Evidence:**

Yes.

**Essential References Not Discussed:**

The reviewer is not aware of missing references.

**Experimental Designs Or Analyses:**

The experimental designs and analyses are reasonable.

**Methods And Evaluation Criteria:**

The methods and evaluation are reasonable.

**Other Comments Or Suggestions:**

1. Equation (3) is confusing, as the parameter k is not shown in the righthand side.
2. $S_i$ appears in line 289 - left column is duplicated as $S_i$ in equation 6.

**Other Strengths And Weaknesses:**

Strength:
1. The paper is well-written, with clear structure and illustrations.

Weakness:
1. The labels from the viewAnnotator come from large vision language models, which can not ensure the quality for the training of viewSelector.

**Questions For Authors:**

1. How do you evaluate/ensure the quality of $S_i$ generated by the LVLM (equation 6)?
2. Does $S_i$ appears in line 289 - left column means the $\hat{S_i}$ in equation 7? Duplicated definition of $S_i$ is confusing.
3. For equation (12), why do you directly add these two distances together? Are they within the similar scale? Have you conducted experiments to test whether the ratio between these two would affect the performance?
4. The conclusion for viewNMS Thresholds part (line 433-left column) is unclear. Search for parameter pairs is time-consuming. Is there any insight for how to choose appropriate thresholds if we are going to apply this pipeline on new datasets?
5. Table S3 indicates the backbone matters a lot in such pipeline. Why llava-OV is doing much better? When using llava-next the performance is worse than bridge-3d.

The reviewer may adjust the final rating after rebuttal based on the clarifications from the authors.

**Relation To Broader Scientific Literature:**

This paper extends the field of 3D question answering by introducing a pipeline which only uses 2D vision language models. The proposed cdViews pipeline is could also be applied to other downstream 3D understanding tasks.

**Theoretical Claims:**

No theoretical claims and proofs involved.

---

> ### Author Rebuttal · Authors · 2025-04-01
>
> ### **1. “... ensure the quality for the training of viewSelector.”**
>
> We agree that using LVLMs alone for annotation may lead to unreliable views.
>
> To address this, viewAnnotator is designed to capture informative views beyond simply image matching. Specifically, we incorporate a step-by-step system prompt (shown in [Figure R1](https://anonymous.4open.science/r/icml_rebuttal-35FF/system_prompt_with_caption.pdf)) in the View Matching process. This prompt ensures that all key objects, attributes, and spatial relationships in the caption align with the image, reducing ambiguity and improving consistency. Uncertain views are explicitly excluded, enhancing the robustness of the annotation process. Additional examples of positive and negative views are shown in [Figure R2](https://anonymous.4open.science/r/icml_rebuttal-35FF/positive_negative_views_with_caption.pdf).
>
> To further validate the reliability of the positive views, we conducted a human evaluation:
>
> - We randomly selected 50 QA pairs with their associated positive views.
> - Three human evaluators assessed whether each view could answer the question.
>
> Their accuracy rates were 96.72%, 94.28%, and 97.56%, confirming that the quality of positive views is sufficient for training.
>
> These details and results will be included in the supplementary document of the revised version.
>
> ---
> ### **2. “Equation (3) is confusing ...”**
> We appreciate the reviewer's detailed comments. In the original Equation (3), the number of sampled views $k$ was not clearly represented on the right-hand side. We will modify the Equation (3) as follows:
> $$
> F_{uniform}(M, k) = \{V_{i_j}\}_{j=1}^k, i_j \sim {Uniform}(1, N).
> $$
>
> ---
> ### **3. “ ... Duplicated definition of $S_i$...”**
> Both instances of $S_i$ on Line 289 (left column) should be updated to $\hat{S}_i$ for consistency with Equation (7). In our notation, $S_i$ denotes the ground-truth label, and $\hat{S}_i$ represents the predicted score from the model. The expressions on Line 289 correspond to model outputs, so the appropriate notation is $\hat{S}_i$.
>
> ---
> ### **4. “... directly add these two distances ...”**
> Yes, the two distances are on comparable scales. Based on statistics from the ScanQA and SQA datasets, position distance ranges from $[0, 14.72)$, and the orientation distance is within $[0, \pi)$. Given their similar scales, they can be combined.
>
> We tested different combination weights on the validation set (Table R1). The results show that viewNMS is fairly robust to weighting variations, with equal weighting (i.e., direct summation) yielding the best performance.
>
> | LLAVA-OV | $D_{pos} \times$ | $D_{ori} \times$ | Threshold $T$ | EM@1       |
> |---------------------------------------------|------------------|------------------|----------------|------------|
> | + $F_{cdViews}$                                          | 0.5              | 1                | 0.5            | 29.7       |
> | + $F_{cdViews}$                                         | 1                | 1                | 0.5            | **30.1**   |
> | + $F_{cdViews}$                                          | 1                | 0.5              | 0.5            | 29.8       |
>
> **Table R1:** *Ablation study on distance weighting in viewNMS. We vary the relative weights of position distance ($D_{pos}$) and orientation distance ($D_{ori}$) while keeping the threshold fixed at 0.5. Equal weighting (1:1) achieves the best EM@1 performance.*
>
> ---
> ### **5. “... viewNMS Thresholds part ...”**
> We clarify that viewNMS does not require searching over parameter pairs. As noted in our response to the previous question (regarding Equation (12)), the position and orientation distances are directly combined with equal weights, so only a single threshold needs to be selected.
>
> This threshold is chosen based on the validation performance. As shown in the line chart in Figure 7 of the manuscript, a threshold of 0.5 yields the best result.
>
> ---
> ### **6. “... the backbone matters a lot ...”**
> Our 3D-QA pipeline consists of three main steps: view selection, feeding the selected view into a 2D LVLM, and generating an answer from the LVLM. Our main contribution is the whole pipeline of using 2D LVLM for 3D-QA. Our add-on contribution is proposing cdViews for more efficient view selection in the first step.
>
> Since most network parameters (in our pipeline) reside in the 2D LVLM, the final performance heavily depends on its pre-trained knowledge. Compared to LLaVA-Next and Bridge-3D, LLaVA-OV performs better due to its improved vision-language alignment—enhancing image understanding and generating more accurate, detailed descriptions—achieved through stronger language modeling and high-quality instruction tuning on open-vocabulary tasks.

---

> > ### Comment · Reviewer_pwsQ · 2025-04-02
> >
> > Thanks for the detailed rebuttal. The authors have addressed most of my concerns.
> > I have read the reviews from other reviewers and authors' rebuttal.
> > I consider the proposed method is effective with solid experimental results, bringing new insights (using 2D VLM models) to 3D-VQA tasks. Whether it is promising to follow this path on 3D-VQA tasks could be left to the community to determine in the future.
> > Thus I would like to keep my original score.

---

### Official Review · Reviewer_rVhy · 2025-03-15

**Overall Recommendation:** 3

**Summary:**

This paper introduces cdViews, a zero-shot method for 3D question answering that avoids fine-tuning large vision-language models (LVLMs). Initially, viewSelector is employed to automatically select the most relevant views based on the input question. Then, viewNMS enhances diversity by eliminating redundant views, determined through their spatial overlap. Finally, the selected views, along with the question, are fed into a pre-trained 2D LVLM (LLAVA-OV) to generate the answer. Experimental results demonstrate cdViews' state-of-the-art performance on the ScanQA and SQA benchmarks.

**Claims And Evidence:**

1. The claim of zero-shot is questionable. The viewSelector requires training on data from existing 3D QA datasets (ScanQA and SQA3D), which may limit its generalization to other 3D QA datasets, particularly when the 3D scenes are not from ScanNet. In such cases, the domain of 2D view images would differ, potentially affecting performance.

**Essential References Not Discussed:**

N/A

**Experimental Designs Or Analyses:**

1. An ablation study is missing to evaluate the effectiveness of viewSelector. For instance, comparing "Image Retrieval" + viewNMS with cdViews (viewSelector + viewNMS) would help quantify the improvement brought by viewSelector over "Image Retrieval." I believe that the combination of "Image Retrieval" and viewNMS could yield comparable performance, and this is a true zero-shot method.

**Methods And Evaluation Criteria:**

1. The viewAnnotator relies on an LVLM, which does not guarantee the accuracy of the annotated positive views. Further analysis is required to assess its reliability.

**Other Comments Or Suggestions:**

Typos:
Line 241: "... the zero-shot 3Q-QA..." (should it be 3D-QA?)

**Other Strengths And Weaknesses:**

N/A

**Questions For Authors:**

N/A

**Relation To Broader Scientific Literature:**

This work is related to 3D question answering and zero-shot learning, offering some insights, such as selecting critical 2D views and using pre-trained 2D LVLMs, for future exploration of zero-shot 3D tasks.

**Theoretical Claims:**

N/A

---

> ### Author Rebuttal · Authors · 2025-03-31
>
> ### **1. “The claim of zero-shot is questionable”**
> Sorry for the confusion caused. For our best method LLAVA-OV + $F_{cdViews}$, the term `zero-shot` could be more precisely scoped: we will revise Lines 153–156 to state it as `zero-shot 2D LVLM inference` (rather than fully zero-shot).
>
> We would like to clarify that our goal in this paper is to make use of 2D LVLMs (which are much better trained than 3D ones) for free, i.e., to adapt them for 3D tasks without any large-scale 3D pre-training or fine-tuning.
> To this end, we proposed two pure no-training methods denoted as LLAVA-OV + $F_{uniform}$ and + $F_{retrieval}$,
> and one with small-scale training (on the viewSelector) denoted as + $F_{cdViews}$.
> During inference, 3D-QA is conducted by feeding the uniformly-sampled / retrieved / selected 2D views and the question into LLAVA-OV, and LLAVA-OV outputs the answer based solely on its knowledge learned from large-scale 2D pre-training.
>
> In the paper, we will revise the motivation paragraph of the Introduction section to make these clearer.
>
> ---
> ### **2. “... which may limit its generalization ...  ”**
> We address this question from two perspectives: the generalization of our 3D-QA pipeline and the trainable module viewSelector.
>
> First, our pipeline leverages 2D LVLMs for 3D tasks without large-scale 3D-language pretraining. It supports three view selection methods: viewSelector, random sampling, and image retrieval. The latter two require no training (assuming minimal domain gap between retrieval models' training data and 3D-QA data). This ensures flexibility and broad generalization of the whole pipeline.
>
> Second, viewSelector is a lightweight trainable module that easily adapts to another similar task with small-scale 3D training data. Compared to large-scale 3D-language pretraining, fine-tuning viewSelector offers a practical way to enable the generalization of 2D LVLMs to 3D tasks.
>
> ---
> ### **3. “... the accuracy of the annotated positive views ...”**
> We agree that using LVLMs alone for annotation may lead to unreliable views.
>
> To address this, viewAnnotator is designed to capture informative views beyond simply image matching. Specifically, we incorporate a step-by-step system prompt (shown in [Figure R1](https://anonymous.4open.science/r/icml_rebuttal-35FF/system_prompt_with_caption.pdf)) in the View Matching process. This prompt ensures that all key objects, attributes, and spatial relationships in the caption align with the image, reducing ambiguity and improving consistency. Uncertain views are explicitly excluded, enhancing the robustness of the annotation process. Additional examples of positive and negative views are shown in [Figure R2](https://anonymous.4open.science/r/icml_rebuttal-35FF/positive_negative_views_with_caption.pdf).
>
> To further validate the reliability of the positive views, we conducted a human evaluation:
> -  We randomly selected 50 QA pairs with their associated positive views.
> -  Three human evaluators assessed whether each view could answer the question.
>
> Their accuracy rates were 96.72%, 94.28%, and 97.56%, confirming that the quality of positive views is sufficient for training.
>
> These details and results will be included in the supplementary document of the revised version.
>
> ---
> ### **4. “An ablation study is missing...”**
> As suggested, we performed ablation comparing the retrieval + viewNMS and cdViews and show the results in Table R1. Combining image retrieval with viewNMS reduces the number of input views due to redundancy removal, but it brings very marginal performance improvement (+0.1%).
>
> The reason is that viewNMS prioritizes diversity, but the overall performance still depends on whether the selected views contain the necessary visual evidence for the question. This highlights a key difference between image retrieval and viewSelector--while image retrieval focuses on matching text and images, viewSelector is explicitly trained to find views that are most important for reasoning.
>
> | LLAVA-OV + Method                  | Image Retrieval | viewSelector | viewNMS | Best EM@1 | Optimal k |
> |-----------------------------------|-----------------|----------------|-----------|-----------|-------------|
> | $F_{retrieval}$  | ✅              | -              | -         | 29.1      | 17          |
> | $F_{retrieval}$  | ✅              | -              | ✅        | 29.2      | 9           |
> | $F_{cdViews}$  | -               | ✅             | -         | 29.7      | 17          |
> | $F_{cdViews}$  | -               | ✅             | ✅        | **30.1**  | **9**       |
>
> **Table R1:** *Comparison of image retrieval and cdViews with and without viewNMS on the ScanQA validation set. The best EM@1 scores are reported with the corresponding optimal $k$.*
>
> ---
> ### **5. “Typos: ...”**
> We thank the reviewer for pointing this out. We will revise it.

---

> > ### Comment · Reviewer_rVhy · 2025-04-02
> >
> > The authors' response has addressed most of my concerns. I would like to increase my rating to 3 -- weak accept.

---

### Decision · Program_Chairs · 2025-05-01

**Decision:**

Accept (poster)

**Comment:**

Almost all reviewers agree on this paper (but not the rating).

- The question: "Do 2D VLMs outperform 3D VLMs in spatial reasoning" seems intriguing.
- The evidence provided in the paper seems incomplete.

The AC agrees with this assessment. Given the additional evidence provided in the rebuttal, the paper seems above bar. While it is not the final word in the discussion weather to 3D or not to 3D, it does provide value to the community.
The AC recommends the authors to include the reviewers feedback in the final version of the paper.